# Alternative Protein-Based Meat and Fish Analogs by Conventional and Novel Processing Technologies: A Systematic Review and Bibliometric Analysis

**DOI:** 10.3390/foods14030498

**Published:** 2025-02-04

**Authors:** Buse N. Gürbüz, Lorenzo M. Pastrana, Ricardo N. Pereira, Miguel A. Cerqueira

**Affiliations:** 1International Iberian Nanotechnology Laboratory, Av. Mestre José Veiga, 4715-330 Braga, Portugal; buse.gurbuz@inl.int (B.N.G.); lorenzo.pastrana@inl.int (L.M.P.); 2Centre of Biological Engineering, Minho University, 4710-057 Braga, Portugal; rpereira@ceb.uminho.pt; 3LABBELS—Associate Laboratory, Braga/Guimarães, Portugal

**Keywords:** alternative proteins, meat analogs, fish analogs, extrusion, 3D printing

## Abstract

This study aimed to explore the extent of research on developing meat and fish analogs using alternative proteins. It examined the novel and conventional technologies employed to produce these analogs and identified the primary alternative proteins that were used in their production through a systematic literature review (SLR) using Preferred Reporting Items for Systematic Reviews and Meta-Analyses (PRISMA) and bibliometric analysis. The SLR resulted in 46 and 13 meat and fish analog records, respectively, according to defined selection and exclusion criteria. Meat analogs are mainly produced using extrusion, followed by the novel 3D printing and mixing technology. Additionally, fish analogs are mainly produced by mixing and 3D printing. Meat analogs are mainly produced from pulses, followed by cereal, fungi, microalgae, other sources, and insects. Similarly, pulse proteins were the most used alternative protein source for the fish analogs, followed by macro- and microalgae, plant, cereal, and fungal proteins. According to keyword analysis, rheological and textural properties are essential for meat and fish analogs. This review provides up-to-date information to clarify the critical role of alternative proteins and the utilization of novel technologies in the production of meat and fish analogs. It also gives essential insights into the expected increase in studies to determine sustainability and overcome challenges related to textural, sensorial, and nutritional properties.

## 1. Introduction

Changes in the environment and climatic conditions have led to significant alterations in food systems, risking food security in the future. In addition, currently, there is an increased importance on food quality and better nutrition, and a growing interest in more sustainable consumption to decrease the environmental impact of newly developed foods [1,2]. Meat plays an important role in human growth and development due to its rich nutritional content in macro- and micronutrients (e.g., protein, iron, vitamin B12) as well as saturated fatty acids [3]. However, it has been reported that more attention should be given to the consumption of red and processed meats to achieve a healthy and environmentally friendly diet [4] due to saturated fat content and the release of increased quantities of greenhouse gases (±1200 g CO_2_-eq/serving) during production. It was reported that one of the main causes of cardiovascular diseases (CVDs) is related to high consumption of saturated fats [5], and CVDs caused the death of more than 18 million individuals around the world in 2019 [6].

The high nutritional value and balanced fatty acid profile of fish products encourage the consumers to increase consumption of fish and seafood products [7], as the data of the Food and Agriculture Organization showed that between 2010 and 2021, the consumption of fish and seafood products per capita increased until 2019 [8]. However, the elevated consumption of fish and seafood products may cause a negative environmental impact on the ecosystem due to overfishing of marine resources. This can lead to low biodiversity in the oceans [9] and a negative health impact due to elevated large-scale marine farming, resulting in an increase number of fish diseases [10]. Thus, growing concern about environmental preservation and animal welfare has led consumers to search for diets or food products that provide similar nutritional composition but show lower negative environmental impact.

Plant-based proteins have commonly been used in food formulations to mimic meat since the early 1970s [11]. However, the demands of consumers for food personalization, mainly for different flavors and textures, have increased. Additionally, increased concerns about health and the environment have led to searching for different alternative protein sources [12]. Alternative protein is defined as proteins that are produced from plant or animal cells or via fermentation. They can present a low environmental impact while increasing or maintaining the sensory, functional, and textural properties of the product where they are incorporated [13]. Besides these properties, alternative proteins also offer to provide the required protein and energy quantities at the same or less cost in less developed countries. These proteins are produced from different sources, such as plants, insects, and microbial sources (e.g., algae, filamentous fungi, and yeast) [14]. Some of the most used plant alternative proteins with high protein content are pulses, including lentil, pea, chickpea, mung bean, and soy, which also have a significant dietary fiber and mineral content, and a low saturated fat content [15]. In addition, they are presented as sustainable ingredients due to improved nitrogen fixation during their production [16], which turns them into a feasible option to be used in food analogs as protein sources. Additionally, bioactive molecules, such as lectins/hemagglutinins and enzyme inhibitors, of the pulse proteins might reduce the risk of obesity and CVDs [17]. However, these proteins might show some allergenic properties because of the existence of compounds such as α-conglutins (legume-like) and β-conglutins (vicilin-like) [18]. Similarly, single-cell protein from fungi (i.e., mycoprotein), is naturally rich in RNA-derived nucleotides and might show allergenic properties, like any other protein source, but they also show high content in protein, dietary fiber, vitamins, and minerals, and low content in saturated fat, and the RNA content can be reduced through heat treatment during production [19,20]. Additionally, it was reported that the consumption of mycoprotein at certain levels is capable of reducing cholesterol levels, increasing muscle protein synthesis and, less conclusively, controlling glucose and insulin levels [20]. Furthermore, another alternative protein source is microalgae, which includes *Chlorella* spp. (unicellular) and *Arthrospira* spp. (multicellular cyanobacteria). One of the most consumed and commercialized algae proteins is produced from the filamentous cyanobacteria *Arthrospira* spp. [21]. They can be produced in open ponds or closed bioreactors, with closed bioreactors having been shown to be more environmentally friendly than open ponds [22]. *Arthrospira* spp. shows increased nutritional composition in protein, vitamins, fatty acids, minerals, and phytonutrients such as chlorophyll-a and phycocyanin, thus providing an improved effect on the nervous system, gut, nephrology, stem cells, ophthalmic system, immune system (e.g., increased antioxidant activity, anti-inflammatory properties), cardiovascular system (e.g., decreased platelet aggregation and blood pressure), oncology (e.g., decreased oxidative stress), and allergies, showing an antiallergenic effect [23]. However, increased quantities of *Arthrospira* spp. in the food matrix might negatively impact sensory properties such as the appearance, taste, and odor of the food product [24]. This can be overcome by using processing technologies such as pulse electric fields and high-pressure homogenization (HPH) [25].

Increased health concerns caused a decrease in meat consumption, which can be explained by consumers’ growing interest in meat analogs to improve their diet and the future of the planet. Meat analogs mainly comprise vegetable-based products; however, due to the recent innovative developments in food science and technology, cultured meat and edible insects are also accepted as meat analogs [26]. Moreover, regarding technological challenges, cultured meat and algae are determined to be highly challenging, as they have recently been used as meat analogs, and they require complicated high-technology processing steps to be included in meat analogs. Additionally, these challenges were identified at a moderate level for processed insects and a low level for pulses and whole insects. Besides their high and moderate level of technological challenges, meat analogs based on cultured meat, algae, and insects require further safety tests due to legal and institutional framework aspects [27]. Fish analogs have also emerged to meet the nutritional needs of consumers with a more sustainable approach. They are mainly composed of plant-based products; however, the increased interest of consumers led food scientists to develop more and new formulations from alternative proteins using novel technologies such as 3D printing [10]. However, meat and fish products can show differences in physicochemical, functional, nutritional, sensorial, and textural aspects. These same differences can be expected from meat and fish analogs [28]. For instance, although meat and fish products are rich in heme iron, meat products contain more heme iron than fish products [29]. Also, fish products show different structural muscle texture due to alternating muscle layers known as myotomes, which are separated from each other and anchored by connective tissue [30]. Additionally, fish analogs are often fortified with ω − 3 fatty acids to produce fatty fish analogs such as salmon fillet [31], whereas meat analogs can be fortified with micronutrients such as vitamin B12, iron, and zinc [32].

The production of meat and fish analogs has been challenging for scientists and industry, as both types of foods (meat and fish) contain different fibrous structures, which are hard to mimic. Several technologies were developed to create fibrous morphology in meat and fish analogs. Bottom-up and top-down [33] approaches have been used to form fibrous structures, with the bottom-up approach based on assembling individual structural elements to create a larger structure, and the top-down approach achieving an anisotropic, fibrous structure by shearing the mixture of proteins and/or polysaccharides. The bottom-up approach includes the methods of electrospinning, wet-spinning, and cell culturing, where the extrusion, mixing proteins and/or hydrocolloids, and 3D food printing are included as examples of top-down approaches [34]. Additionally, the opportunities and challenges of these traditional and new production technologies have been discussed elsewhere [35]. In one review, the authors explained that these production technologies tend to produce food analogs with desirable sensory properties, especially chewiness. Though these technologies are capable of producing chewy and juicy food analogs, they are not able to create a muscle fiber–myofibril network entirely. Thus, the authors recommended that further studies should focus on novel technologies that mimic capillary systems in muscles to obtain increased chewiness and juiciness characteristics [35].

The objective of this study was to conduct an SLR and bibliometric analysis of meat and fish analogs produced using alternative proteins, and provide a diverse and clear overview of the utilization of alternative proteins from different sources, such as algae, plants, and fungi, to produce meat and fish alternatives using novel and conventional technologies. Therefore, for the literature review of meat and fish analogs, the following research questions were answered: RQ1. What is the scope of research that has been conducted on the development of meat and fish analogs based on alternative proteins? RQ2. What type of novel and conventional technologies have been used to produce alternative protein-based meat and fish analogs? RQ3. What are the main types of alternative proteins that have been used to produce meat and fish analogs?

## 2. Materials and Methods

A systematic literature review was conducted following the Preferred Reporting Items for Systematic Reviews and Meta-Analysis (PRISMA) guidelines [36], using Scopus and Web of Science for papers published in English until May 2024 without specifying the initial date. Table 1 and Table 2 present the keywords and Boolean operators that were used for the research, according to the type of analogs and the databases Scopus and Web of Science, respectively.

The PRISMA [36] flow diagram was used to reveal the number of records identified, and the selection and rejection process of the papers that were found. The key criteria of the selection process were defined as follows: S1. Articles should be written in the English language and published in peer-reviewed journals. S2. Studies should include at least one type of alternative protein to produce meat and fish analogs. S3. Studies should identify the type of technology that was used to produce meat and fish analogs. S4. Studies should identify the final type of meat or fish analog.

On the other hand, exclusion criteria were based on the following: E1. Review articles, E2. No access to the full paper, E3. Analysis of commercially available food analogs without presenting the production method, E4. Consumer sciences studies with commercially available food analogs, E5. No final food analog production.

Thus, the main objective of the exclusion criteria to answer the question was whether the article includes the production process of the food analog using alternative protein. The detailed selection process of meat and fish analogs of this study is presented in Figure 1 and Figure 2, respectively.

Data were collected by a reviewer from selected databases for each type of food analog using the indicated search strings (Table 1 and Table 2). All data were imported into Mendeley Reference Manager (version 2.116.0) for removal of duplicates. After the duplicate removal process, records were compiled into a data sheet with the following information: authors, title, year, source title, citation count, DOI, abstract, author keywords, index keywords, and affiliation. Then, the records were screened by the reviewer based on their title, keyword, and abstract, considering criteria for selection and exclusion. Next, eligible records were subjected to more detailed assessment by reading the entire publication. Records were included for the SLR upon agreement of the reviewers, considering the selection and exclusion criteria.

Selected relevant records were analyzed using bibliometric methods such as co-occurrence of keywords and co-authorship analysis, which is an important technique for determining the patterns in development or detecting the direction of future research [37]. Thus, network analysis based on co-occurrence of keywords and co-authorship relations was conducted using “visualization of similarities”—VOSviewer software (version 1.6.20) to determine the important keywords and the most significant authors in each research field selected, respectively. Additionally, the keywords needed to be merged due to spelling differences, synonyms, and abbreviated terms using thesaurus files. For example, the terms “3-D printing”, “3D-printing”, and “three-dimensional printing” were merged to “3D printing”. Nevertheless, despite these limitations, this work provides a comprehensive analysis of the development of meat and fish analogs using alternative proteins and different technologies. Normalization of the data is an important step to obtaining more accurate visualization. The normalization methods include association strength, fractionalization, and LinLog/modularity. Among these methods, association strength and fractionalization use similarity measures [38], and LinLog/modularity applies a modularity measure [39,40,41]. In this study, LinLog/modularity was chosen as the normalization method to minimize the distance between connected nodes, as this technique allows the node position to be determined by edge density while ignoring path length, which allows for the superposition of nodes with high collinearity. With this method, the data was normalized using LinLog/modularity and it can be visualized as clusters (a set of closely related nodes) organized in subclusters, which enables more comprehensive data visualization.

## 3. Results

### 3.1. Publication Tendencies for Meat and Fish Analogs

The publication year of the selected articles, according to the criteria established previously (S1, S2, S3, and S4), varied between 2003 and May 2024 for meat analogs, and between 2022 and May 2024 for fish analogs (Figure 3).

According to the results of the search string used in this research, it seems that the research about the production of meat analogs based on alternative proteins appeared in the early 2000s and increased during the consecutive years. It is possible to observe an increase in the number of publications starting from 2021. The peak in the number of publications was in 2023, with 18 records. Additionally, it seems that the production of fish analogs from alternative proteins is still a new topic in the food sciences. It is possible to detect a significant increase in the quantity of scientific production within only one year (2022–2023). This research was conducted in May 2024; thus, it includes the papers available by 31 May 2024. Figure 3 shows that in 2024, eight papers for meat analogs and two papers for fish analogs were included. However, according to the increased interest in meat and fish analogs to develop new products with higher nutritional quality and decreased environmental impact, the number of publications is expected to continue to increase in upcoming years.

### 3.2. Main Findings of the SLR for Meat Analogs

All selected articles for meat analogs are compiled in Table 3 and divided into five categories: (i) type of analog, (ii) protein type, (iii) composition of the meat analog, (iv) tested parameters, and (v) technology used to produce the meat analog (Table 3).

The diversified categories of the alternative proteins used in the formulations of meat analogs are shown in Figure 4.

It was observed that pulse proteins were used in 43 documents with a combination of one or more of the same protein or a different type of protein. The results showed that pulse proteins have been used as the main protein source, followed by cereals, fungi, microalgae, insects, and proteins from other sources. In this study, pulse proteins included chickpea, CP, faba bean, GG, HG, lentil, lupin, mung bean, pea, soy, and texturized pea and soy protein. Cereal proteins included wheat gluten and rice protein, while fungal proteins included mycelium (*Pleurotus eryngii*), and mycoprotein (*Fusarium venenatum*, *Neurospora intermedia*, *Penicillium limosum*). *Arthrospira* spp., yellow chlorella (*Chlorella protothecoides*), *Auxenochlorella protothecoides*, and *Haematococcus pluvialis* protein were included in microalgae proteins, and mealworm (*Tenebrio molitor*), cricket (*Gryllus bimaculatus*), black soldier fly (*Hermetia illucens*) larvae, and lesser mealworm (*Alphitobius diaperinus*) were included in insect proteins. Additionally, other proteins included rapeseed (oilseed), canola (oilseed), duckweed (flowering plant), potato (tuber), microbial single cell, and mushroom protein (reishi (*Ganoderma lucidum*), saffron milk cap (*Lactarius deliciosus*), and oyster mushrooms (*Pleurotus ostreatus*)).

The wide utilization of pulse proteins in meat analogs can be related to their ability to form fibrous and structural matrices through improved gelling properties, while showing feasible extraction yields with low cost. This review showed that the most common pulse protein to produce meat analogs was soy protein, followed by pea protein. Pulse proteins show high nutritional value, with balanced amino acid composition, and their protein content varies between 17 and 30% depending on the type [88]. Additionally, they may contain bioactive constituents and enzyme inhibitors, which can help to reduce serum glucose levels and reduce the risk of obesity [17]. On the other hand, in this study, the second most used protein type was from cereals, which contain 7–15% protein and show several health benefits, such as reduced risk of obesity, effectiveness against hypercholesterolemia, and beneficial effects for the heart [89]. The techno-functional properties, such as solubility, emulsifying, and water- and oil-holding capacity, of cereal proteins depend on the intrinsic (e.g., amino acid composition, structure) and extrinsic (e.g., pH, temperature) characteristics, and can be modified through physical, chemical, and biological methods to improve their incorporation into matrices [90]. Thus, with increased sustainability concerns, the utilization of cereal proteins in the food industry is rising. Additionally, it seems that the utilization of alternative proteins such as fungi and microalgae-based on the composition of meat analogs is rising. They are considered promising sustainable nutritious proteins because they can contain all essential amino acids, depending on the strain, and their protein content can vary between 10 and 45% and between 1 and 71%, respectively [22,91].

The records included in this study showed that different protein sources were used to produce meat analogs in different years (Figure 5).

For instance, fungal proteins such as mycoprotein were the oldest protein source to produce meat analogs, as their first utilization was in 2003, followed by pulse and cereal proteins, which were first used in 2011. Additionally, proteins from microalgae and insects were started to be used to produce meat analogs in 2018. Furthermore, other proteins, such as canola, duckweed, or potato, have been used in meat analog production since 2023.

Pulse proteins are an important protein source due to their functionality and balanced protein content compared to real meat products. Thus, they are applied widely to produce more sustainable meat products [92]. The utilization of mycoprotein since the early 2000s may be related to its enhanced nutritional properties and filamentous hyphae structure, which allows for fibrous bundles mimicking meat texture to be obtained. However, it seems that the studies that evaluated mycoprotein utilization to produce meat analogs were interrupted for 10 years (between 2011 and 2021). This interruption may be related to its safety for human consumption, as its slow growth can cause contamination depending on the strain, and its high nucleic acid composition can cause urolithiasis by increasing the uric acid concentration in the blood [93]. Thus, to overcome these concerns, several studies were conducted to determine mainly its toxicity due to the possible existence of mycotoxins, as well as the allergenic effects. These studies showed that mycoproteins can be considered safe to consume depending on the strain [91]. The utilization of insect- and microalgae-based protein is considered a sustainable approach to producing meat analogs with well-balanced essential amino acid composition depending on the species [22,94]; thus, their incorporation into food matrices such as meat is increasing to determine their feasibility and determine the main challenges. Additionally, cereal proteins seem to be gaining more interest in terms of meeting the demands for both healthier and greener protein sources. Similarly, other proteins, such as canola or rapeseed, contain 17–26% protein, and generally show high functional properties such as solubility and emulsification. Although they can show unfavorable sensory characteristics, their versatility, economical availability, and sustainability allow them to be incorporated into different food matrices, from baked goods to meat. Thus, studies to determine and improve their negative properties are increasing [95,96].

The extrusion process, mostly high-moisture extrusion, was used in 17 studies (37%), followed by the novel technology of 3D printing with 16 studies (35%), and mixing with 13 studies (28%). Three-dimensional printing allows for the food product to be personalized, where the shape, textural, and rheological properties can be controlled by parameters such as nozzle size, printing, and motor speed [97]. Besides textural and rheological properties, personalization of the nutritional composition can be achieved through the addition of established components and quantities to the ink composition of the food product to be printed. Additionally, 3D printing provides a sustainable approach, as it allows for the production of food products using raw materials that show low environmental impact. Moreover, 3D printing contributes to decreased food waste because it allows for the exact amount of desired food products to be produced. [98]. Although currently, 3D printing shows several challenges from a technological (e.g., speed and large-scale production capacity), economic (e.g., high initial investment), and consumer perception (e.g., skepticism by consumers) point of view [99], according to our study, 3D printing seems to be a promising production technology for meat analogs in the future, as its application in the food area is relatively recent [100].

### 3.3. Overview of the Studies Selected for Meat Analogs

#### 3.3.1. Extrusion

The selected articles show that studies on alternative protein-based meat analogs started in the early 2000s. For instance, Kim et al. [42] investigated the influence of cooling and rehydration methods on an HMMA based on pea, lentil, and faba bean proteins using four different cooling methods (at room temperature (25 °C) in air, in water, in a 2% brine solution, and in a 4% brine solution) and three different rehydration treatments (soaking, warm soaking, and boiling). The authors reported that the water solubility index decreased when the HMMA based on pulse protein was cooled in a 2% brine solution and rehydrated with boiling water. In the study by Fu et al. [43], the effect of three different polysaccharides (kc, curdlan, and potato starch) on the textural and structural properties of high-moisture extrudates like a meat analog based on pea protein was studied. The study showed that the incorporation of curdlan and kc increased the hardness and chewiness, while the incorporation of potato starch decreased the same textural properties. Additionally, the study revealed that the incorporation of kc led to sharp structures, the incorporation of curdlan led to short and flat structures, and the incorporation of potato starch led to sharp and flat fibrous structures. The authors reported that the addition of polysaccharides had a high impact on the textural and structural properties of the meat analogs, as excessive quantity led to poor textural and sensory properties. Usman et al. [44] prepared an HMMA with pulse proteins (lentil and pea) and determined the influence of the complex effect of germination and extrusion processing on sensory characteristics. The study showed that the extrusion process decreased the odor, and both the extrusion and germination processes changed the odor of the meat analogs by affecting the flavor compounds for both pulse protein types. Additionally, lentil-based meat analogs showed a dark-brown color and low chewiness, whereas pea-based meat analogs showed a lighter color and high chewiness. The authors reported that the combination of germination and extrusion processes led to meat analogs with increased textural and sensory properties. Barnés-Calle et al. [45] produced a meat analog from pea protein isolate using high-moisture extrusion and evaluated the physicochemical and sensory properties of the HMMA. The authors reported that during the extrusion, the temperature of the extrusion and the water-feeding rate directly affected the moisture content and textural properties of the HMMAs, and they suggested that to obtain textural properties similar to those of real meat, the ideal temperature and water-feeding rate should be between 145 and 165 °C and between 53 and 57%, respectively. Additionally, they reported that, according to the results of the appearance and sensory fibrousness, the most promising HMMA compared to real meat was produced at 165 °C with a 55% water-feeding rate. Guo et al. [46] investigated the effect of germination and the type of pulse protein (pea and lentil protein) on an HMMA. The study showed that the structural and functional properties of both pulse proteins were affected by the germination process, where the protein composition was changed by decreasing the molecular weight of the subunits, and the melting temperature of both pulse proteins was increased. Additionally, it was reported that the melting temperature of the protein affected the final textural properties of the meat analog, with higher melting temperatures usually tending to produce less cohesive, chewy, and gummy HMMAs. Moreover, a higher melting temperature of the proteins often lead to producing HMMAs with a more stable and robust structure because of possible strong intermolecular forces, improved hydrophobic interactions, or an elevated number of ionic bonds in the protein structure. The authors reported that even the extrusion process provided fibrous structure formation, and a minor improvement was observed in the structural profile of the germinated pulse protein-based meat analogs. In the study by Zhang et al. [47], meat analogs were produced from an alternative protein of rapeseed protein mixed with soybean protein in different ratios (0:50 to 50:0 wt%) using low-moisture extrusion. They concluded that the addition of rapeseed protein of up to 20 wt% in the formulations led to reduced elasticity and improved hardness and chewiness, resilience, specific mechanical energy, and mass flow rate. Additionally, the incorporation of rapeseed protein of more than 20 wt% decreased the expansion characteristics, internal pore structure, water absorption rate, and surface brightness while increasing the redness of the surface. Furthermore, they reported that the addition of rapeseed protein in the formulations was positively correlated with the decreased protein denaturation. Thus, they suggested the addition of 10–20 wt% rapeseed protein can improve the physicochemical and structural properties of meat analogs produced using low-moisture extrusion.

Grahl et al. [48] studied the influence of the technical parameters of extrusion methods on the sensory properties of meat analogs based on soy and *Arthrospira* spp. In their study, meat analogs were produced using extrusion with different levels of *Arthrospira* spp. content (10%, 30%, and 50%), temperature (140 °C, 160 °C, and 180 °C), screw speed (600 rpm, 900 rpm, and 1200 rpm), and moisture (57%, 67%, and 77%). They reported that it was possible to produce meat analogs using *Arthrospira* spp.; however, for the higher quantities of *Arthrospira* spp., the meat analog showed a darker color, an intense flavor with earthy notes, and a musty algae odor. Additionally, partial replacement of soy protein with *Arthrospira* spp. was achieved using low moisture, high screw speed, and high temperature during the extrusion, which also provided firm and fibrous meat analogs. Palanisamy et al. [49] investigated the physicochemical and nutritional properties of meat analogs based on *Arthrospira* spp. and lupine protein mixtures produced using high-moisture extrusion with variable working parameters for *Arthrospira* spp. concentration (15, 30, and 50%), temperature (145, 160, and 175 °C), water feed (50, 55, and 60%), and screw speed (500, 800, and 1200 rpm). The authors reported that *Arthrospira* spp. content of up to 50% in the mixture was capable of producing a meat analog. Additionally, texture, cooking yield, expressible moisture, antioxidant activity, and in vitro protein digestibility of the meat analogs were improved by modifying the extrusion parameters (i.e., temperature, water feed, and screw speed). Caporgno et al. [50] prepared the meat analogs based on microalgae (yellow, heterotrophically cultivated *Auxenochlorella protothecoides*) in combination with soy protein concentrate using dry extrusion. Afterwards, they applied high-moisture extrusion cooking to dry-extruded meat analogs and evaluated the vitamin composition, textural properties, and microstructural properties. The authors reported that the addition of microalgae improved the nutritional quality of the meat analog by providing high levels of vitamins B and E. Additionally, the incorporation of the microalgae in combination with soy protein concentrate in different moisture contents provided different fibrous structure formations, textural properties, and colors. Thus, the authors suggested that to obtain a meat analog similar to real meat in terms of fibrous structure and mechanical texture, the most promising microalgae concentration and moisture content should be 30 wt% and 60%, respectively. Xia et al. [51] evaluated the structural and rheological properties of meat analogs prepared with different proportions of HPR and pea protein using high-moisture extrusion. The authors reported that the addition of HPR improved the appearance and changed the rheological properties by increasing the fluidity. Additionally, the study showed that an addition of 10% HPR provided the best fibrous degree in the meat analogs. In the study by Gol et al. [52], they prepared extruded meat analogs with pea protein and modified microalgae (cell-disrupted *Chlorella vulgaris*) to evaluate the effect of the modified microalgae on the textural properties of meat analogs. The study showed that the incorporation of 10 wt% modified microalgae into pea protein-based meat analogs using high-moisture extrusion was successfully achieved. The authors reported that the incorporation of the modified microalgae did not affect the fibrous structure, hardness, appearance, or anisotropy of the meat analogs. In the study by Liu et al. [53], the effect of microalgae (*Haematococcus pluvialis*) addition at different concentrations (1, 3, 5, and 7%) on extruded gluten-based soybean and wheat meat analogs was evaluated. The study revealed that the addition of microalgae improved the visual appearance, especially the color, and affected the odor of the meat analogs due to the existence of fishy compounds. The authors reported that meat analog showed pseudoplastic flow properties, and the addition of microalgae changed the rheological behavior.

For instance, Miri et al. [54] used mycoprotein as an alternative protein to produce meat analogs to visualize the morphology and hyphal structure. In their study, mycoprotein paste was prepared by staining the mycoprotein fermentation broth and filtering under vacuum to form a paste. A sausage-like meat analog was produced using mycoprotein paste through the extrusion method. The fibrous structure of the native mycoprotein paste and sausage-like meat analog was determined through fluorescence microscopy. The results showed that the fiber orientation in native pastes was random in all directions (isotropic), with the extruded mycoprotein pastes showing altered fiber orientation. Thus, the authors concluded that the method of processing the paste could alter the fiber orientation in mycoprotein paste-based sausage, and the fluorescence microscopy technique was able to visualize and identify the changes in the fiber orientation. Similarly, Miri et al. [55] investigated the effects of extrusion and squeeze flow processing on the microstructure of meat analogs based on mycoprotein. In their study, two different samples were prepared for the flow-processing test: mycoprotein native paste stained with Calcofluor White M2R and a suspension of stained mycoprotein fibers in golden syrup. Additionally, for the extrusion-processing tests, the samples were prepared with different die diameters, as the mycoprotein paste was forced through the dye. The study revealed that processing methods, extrusion, and squeeze flow affected the filamentous microstructure of the mycoprotein. In the extrusion method, fiber alignment was mainly influenced by die diameter, with a decrease in the die diameter leading to a bigger change in fiber orientation. In the squeeze flow-processing test, the fiber alignment of the suspension of stained mycoprotein fibers in golden syrup was affected by the processing method, with the fiber alignment occurring in the radial direction within the boundary layer, and a higher degree of fiber alignment occurring in a direction normal to the radius in the middle layer. Thus, their work showed that the processing method influences fiber orientation in meat analogs, which is considered an important parameter for these products. Mandliya et al. [56] prepared low-moisture meat analogs based on pea protein isolate with mycelium incorporated to determine the effects of the mycelium on physicochemical and microstructural characteristics. They developed five formulations where the concentrations of the mycelium were 0, 10, 20, 30, and 40 wt%, and meat analogs were produced using low-moisture extrusion. The authors reported that the incorporation of a mycelium of up to 30 wt% provided a better microstructure and secondary structure, and increased the water solubility index, water absorption capacity, oil absorption capacity, water-holding capacity, and volumetric expansion ratio. In the study by Zhang et al. [57], the meat analog was prepared using mycoprotein from *Penicillium limosum* and pea protein isolate using high-moisture extrusion. They evaluated the impact of alternative protein utilization on the structural and functional properties of the HMMA. They pointed out that mycoprotein from *Penicillium limosum* was safe to use in the formulation of meat analogs. Additionally, the addition of 5 wt% in the HMMA improved the viscosity, chewiness, and protein digestibility, whereas the increased addition of mycoprotein led to a weak degree of fibrousness and low protein digestibility. Moreover, according to the sensory evaluations, the HMMA with an incorporation of 5 wt% mycoprotein showed the highest overall liking scores. Thus, the authors reported that the incorporation of 5 wt% mycoprotein from *Penicillium limosum* into an HMMA can improve the structural, functional, sensory, and nutritional properties of the HMMA.

In the study by Smetana et al. [58], meat analogs based on different proportions of insect biomass and soy protein were produced using high-moisture extrusion. The authors reported that the main factors that affected the textural properties of the meat analogs were sample composition and water content. The results of textural properties and SEM showed that a mixture of protein of up to 40% insect biomass with a lower water content during the extrusion process provided increased fiber formation and improved cutting strength.

#### 3.3.2. Mixing

In the study by Penchalaraju and Bosco [59], meatball analogs were developed using GG, HG, and CP, and the functional properties of the pulse proteins were studied. The samples were prepared with three different ratios of the pulse proteins (20:20:20; 30:15:15; 15:20:15; GG:HG:CP) and mixed with spice mix, meat masala, salt, corn flour, black pepper, ginger garlic paste, chopped onions, coriander leaves, baking soda, potato starch, and beet root before being deep-fried in sunflower oil. The authors reported that GG protein concentrate showed higher gelation capacity, higher protein solubility (at pH 2 and pH 9), higher emulsion capacity and stability, and the highest value for oil absorption capacity. However, the meatball analogs showed lower oil absorption compared to conventional meatballs. Additionally, the meatball analog prepared in the ratio of 20:20:20 showed a higher value of sensory attributes such as appearance, flavor, taste, texture, juiciness, and overall acceptability. Bakhsh et al. [60] studied the physicochemical, textural, and visual characteristics of the meat analogs prepared with natural pigments (anthocyanin, fe-chlorophyll, dilute red, dilute red 2, red color CG2, paprika, monascus color no. 30, red rr, purple grape, cherry red, monascus color 100, red cabbage liquid, red cabbage 100, af beet red 30, grape skin color, and red color pb) and compared them to meat analogs prepared with animal-based pigment (myoglobin). The authors reported that there were no significant differences between the incorporation of natural and animal-based pigments in terms of moisture, crude protein, crude fat, and ash content. However, significant differences were found in textural properties, with the meat analogs with natural pigments showing lower values for hardness, chewiness, and gumminess. Additionally, meat analogs prepared with natural pigments showed higher antioxidant activity; thus, the authors concluded that it was possible to modify the color of the meat analogs using natural pigments. Penchalaraju et al. [61] investigated the morphological, sensory, physicochemical, and textural characteristics of meatball analogs from pulse proteins (GG, HG, and CP with different ratios (20:20:20; 30:15:15; 15:20:15; GG: HG: CP)) and compared them with mutton meatballs (control sample with meat). The study showed that all pulse protein concentrates showed a collapsed and wrinkled surface, which might have resulted from the spray-drying process. Also, samples including HG in their formulation provided better thermal stability due to the high denaturation temperature of HG protein. The authors reported that textural parameters such as hardness, cohesiveness, and adhesiveness, as well as the color and sensory properties of the meat analogs, were altered according to different ratios of the pulse proteins, and all the formulations showed similar results to the control sample. In the study by Peñaranda et al. [62], hamburgers enriched with lucerne, spinach, or chlorella were produced from pea protein (texturized or in powder), and physicochemical characteristics were evaluated. The study showed that hamburgers prepared with texturized pea protein presented higher water retention capacity and lower cooking loss; on the other hand, hamburgers prepared with pea protein powder showed an increased water-holding capacity than proteins denatured during extrusion, which resulted from the native globular structure of the pea protein. Additionally, the color of the meat analogs changed depending on the enrichment source, with the chlorella-enriched hamburgers showing a darker color before and after cooking. The authors reported that hamburgers prepared with texturized protein showed higher values for most of the sensory and textural properties, especially for juiciness, fibrousness, hardness, and chewiness.

In another study, Bakhsh et al. [63] investigated the rheological, textural, sensory, and nutritional properties of meat analogs with different concentrations of microalgae proteins incorporated, such as *Arthrospira* spp. (0.5, 0.7 and 1%), duck weed (0.5, 0.7 and 1%), and yellow chlorella (1, 2 and 3%). The authors reported that the incorporation of microalgae proteins in different ratios modified the textural properties of the meat analogs, with the increased content of the microalgae resulting in increased hardness, gumminess, and chewiness (highest for the yellow chlorella, at 3%). Additionally, sensory parameters did not show significant differences between the microalgae proteins and concentrations, excluding the meat analog with 1% *Arthrospira* spp. Also, the incorporation of microalgae increased the antioxidant activity and showed a heavy-metal-free micronutrient composition for all formulations. Benevides et al. [64] produced a hamburger analog using the microalgae *Arthrospira* spp., lentil protein, and cashew fiber through mixing and evaluated the aroma, flavor, and overall liking, as well as the moisture, protein, lipid, and ash content. The authors reported that the hamburger analog based on *Arthrospira* spp., lentils, and cashew fiber showed a protein content of between 14 and 17% on a dry basis, and high scores for overall liking, aroma, and flavor. They suggested that the production of this hamburger analog seems like a promising alternative for meat analogs, as it showed high scores of overall liking.

Additionally, Kim et al. [65] studied fungal proteins to identify an economically viable industrial bioprocess for developing a method for mycelium production through submerged fermentation, as well as to characterize mycelium to produce low-calorie meat analogs. They prepared two different meat analogs: a soybean-based and a mycelium-based meat analog. The study showed that the utilization of sugar cane extract, sodium nitrate, and yeast extract for bioprocessing mycelium was economically viable. Also, they reported that it was possible to produce a meat analog from bioprocessed mycelium with improved textural qualities such as hardness, springiness, and chewiness, compared to soybean-based meat analogs. Shahbazpour et al. [66] investigated the effects of the replacement of meat by mycoprotein in cooked sausages on physicochemical, microbial, nutritional, and mechanical characteristics. The mycoprotein sausages were prepared by mixing the mycoprotein, sunflower oil, ice, mixed spices, soy protein isolate, gluten, flour, and salts. After, the mixture was stuffed into impermeable cellulose casings. The authors reported that mycoprotein sausage showed increased nutritional value due to the high value of essential amino acids and low lipid content, which was mainly unsaturated fatty acids. Additionally, the samples showed an absence of foodborne pathogens (i.e., molds, *Salmonella* spp., *Escherichia coli*, and *Staphylococcus aureus*) or less than 10 cfu g^−1^ of foodborne pathogens (i.e., *Bacillus cereus* and *Clostridium perfringens*) after heat treatment. However, textural properties, such as hardness, cohesiveness, gumminess, and springiness, were decreased in mycoprotein sausages due to excessive water and lipid content. Thus, the authors suggested that it was possible to produce sausages with improved nutritional quality using mycoprotein; however, the water and lipid content in the formulation should be decreased to improve the textural properties. In the study by Niimi et al. [67], a minced meat analog was produced using mycoprotein, soy protein, and oatmeal protein through mixing. They evaluated the effect of cooking ability on the sensory characteristics of the minced meat analog served with a complementary matrix of tomato sauce. The authors reported that the cooking ability could impact the liking of the samples depending on the protein type, with the soy protein-based minced meat analog showing the highest scores for liking, and the oatmeal protein-based minced analog showing the lowest scores. Hashempour-Baltork et al. [68] studied the physicochemical and sensory properties of meat analogs (nuggets) based on mycoprotein. The study revealed that nuggets based on mycoprotein showed increased nutritional properties due to high essential amino acids and low lipid content, and indicated the same sensory and textural properties as nuggets made with chicken meat. The authors reported that the utilization of mycoprotein as a meat substitute in nuggets is possible and provides a healthier, more economical and sustainable option for human nutrition. In the study by Wang et al. [69], a meat analog as a composite mycoprotein gel meat was produced using mycoprotein from *Neurospora intermedia* through mixing with soy protein isolate, deionized water, different concentrations of soluble starch, and gluconolactone at varying pH. The authors reported that the strain of *Neurospora intermedia* ZJU-23 was suitable for production of meat analogs. Additionally, the changes in the pH and concentration of soluble starch affected the texture and WHC, where pH 3 and a soluble starch concentration of 6 wt% showed enhanced chewability and gelatinousness, the highest WHC, and limited tensile behavior due to the strong gel structure. The authors reported that the composite mycoprotein gel meat prepared at pH 3 and with a 3 wt% soluble starch concentration showed improved cutting resistance and ductility, mimicking meat tearing. Thus, the authors introduced a cost-effective non-animal excipient fungus strain as a potential protein source to produce meat analogs. Okeudo-Cogan et al. [70] produced meat analogs from mycoprotein, potato protein, FePP, sodium chloride, calcium chloride, and distilled water through mixing. They evaluated the effect of the addition of potato protein, FePP, sodium chloride, and calcium chloride on the appearance, microstructure, and rheological and structural properties of meat analogs. The authors reported that the addition of FePP changed the color of the composite containing only mycoprotein to reddish-brown, reduced the *G’* values, and enhanced the protein–protein aggregation at pH 3. Further, the addition of FePP to the composite containing a combination of mycoprotein and potato protein led to an increase in *G’* values at pH 3 and 7. Moreover, the addition of FePP and calcium chloride together to the composite containing a combination of mycoprotein and potato protein improved the rheological properties and reduced protein aggregation compared to the composite containing mycoprotein only. The authors concluded that potato protein can be used as a binding agent in mycoprotein-based meat analogs; however, the effect of the pH and the addition of cations such as calcium and iron should be studied beforehand.

Miron et al. [71] investigated the effect of insect protein (black soldier fly larvae) incorporation on the textural properties of a soy protein- and vital wheat gluten-based meat analog. The authors reported that the concentration of the insect protein in the mixture was capable of changing the textural properties of the final meat analog due to interactions between different proteins, where the values of the hardness, chewiness, cohesiveness, and springiness were modified by adding 6.7 g/100 g of insect protein to the mixture to obtain a meat analog with similar textural properties to chicken breast.

#### 3.3.3. 3D Printing

Ko et al. [72] produced a meat analog using a coaxial-assisted 3D food printer and investigated the changes in the texture through the inserted fibrous structures using hydrocolloid crosslinking. For the 3D printing of the meat analog, soy protein paste was prepared by dissolving isolated soy protein, potato starch, and xanthan gum in distilled water, and a fiber solution was prepared with dissolving sodium alginate, konjac GM, calcium chloride dehydrate, and potassium chloride in water. The authors reported that it was possible to create a strong and stable 3D structure (i.e., meat analog) by inserting the ionic bond-based hydrocolloids into the center of the protein matrix using the coaxial nozzle-assisted 3D food printer. Additionally, less cooking loss and increased hardness were reported for the hydrocolloid-inserted meat analogs, which presented a similar hardness to beef. Thus, the authors concluded that fiber insertion during the coaxial nozzle-assisted 3D-printing process enabled a textural improvement in the meat analogs. In the study by Shahbazi et al. [101], reduced-fat meat analogs were produced from soy protein isolate and different types of surface-active biopolymers (ethyl cellulose, octenyl succinic anhydride starch, acetylated wheat starch, dodecenyl succinylated inulin) using 3D printing. They evaluated sensory, microstructure, textural, physicochemical, and tribological properties. They reported that the incorporation of biosurfactants in soy protein-based reduced-fat meat analog formulations improved both the formation of protein anisotropic structures and the fibrous degree. Additionally, printed reduced-fat meat analogs containing dodecenyl succinylated inulin and ethyl cellulose as biosurfactants showed finer resolution, compact structure, decreased surface–surface contact, and friction coefficients while improving the lubrication property. Also, the reduced-fat meat analogs containing biosurfactants showed the desired sensory profile, with the authors suggesting that the replacement of oil with biosurfactants improved the resolution and shape-fidelity of the 3D-printed reduced-fat meat analogs. In the study by Leelapunnawut et al. [74], the effects of TGase and kc on 3D-printed meat analogs prepared with pea protein and alginate gel were investigated. The samples were prepared with different concentrations of texture modifiers (TGase and kc) to identify the possible differences in textural and rheological properties. The authors reported that different concentrations of texture modifiers showed no significant differences in rheological properties; however, the hardness of the raw meat analog treated with 0.9 wt% TGase was the highest, while the hardness of the cooked meat analog treated with 0.9 wt% kc was the highest. Thus, the study showed that texture properties of the 3D-printed meat analogs based on pea protein and alginate were affected by the type and the concentration of the texture modifiers. Shahbazi et al. [75] produced 3D-printed meat analogs using soy protein-based Pickering emulsion stabilized by microcrystalline cellulose. They evaluated the printing performance, morphology, and dynamic sensory profile of the 3D-printed meat analogs. They reported that with the increased concentrations of microcrystalline cellulose, the printing performance provided improved layer resolution leading to a high geometrical accuracy. Also, the addition of microcrystalline cellulose provided highly porous structures and improved the temporal perceptions of fibrousness and juiciness. In another study, Wang et al. [76] produced 3D-printed plant-only and plant-based hybrid nugget-shaped meat alternatives based on pea protein. In their study, different nozzle sizes for the 3D printing were tested to compare with the rheological behavior and optimize the formulations. The results showed that, before printing, both plant-only and hybrid meat analogs showed weak gel behavior. Additionally, after printing, the authors reported that a smaller nozzle size provided better 3D shape-forming capacity, which they explained was due to the high shear viscosity from the bigger nozzle. Wen et al. [77] developed meat analog formulations based on mung bean protein, beet red, and xylose using 3D printing, and studied the effect of xylose on the rheological and physicochemical properties of the meat analogs. The authors reported that before printing, the addition of xylose increased the printability of the mixture of the meat analog due to improvement in the shear modulus, and xylose was able to modify the texture by changing interactions between polymers. Additionally, after cooking, the meat analogs showed changes in color, and the authors linked these changes with the Maillard reaction due to xylose. Similarly, in another study, Wen et al. [78] produced meat analogs based on mung bean protein using 3D printing. TGase was added, and its effect on texture, rheological properties, and printability was evaluated. Also, the influence of different cooking methods, such as steaming, microwave, baking, and frying, on the physicochemical and microstructural properties of the meat analogs was studied. The authors reported that TGase was able to modify the rheological and hardness properties of the meat analogs, where 2 wt% TGase provided a smooth surface and relatively high hardness, which led to better printability. Also, the same study showed that different cooking methods led to similar textural properties but different sensory properties, with steaming and microwaving demonstrating a softer mouthfeel, and baking and frying demonstrating a firmer mouthfeel and hard crust. Calton et al. [79] investigated the influence of paste formulation of a meat analog based on pea protein, single-cell protein, and hydrocolloids on structure and texture formation during 3D extrusion printing. The authors reported that 3D printing provided anisotropic textural properties. Also, a higher content of single-cell protein showed a decreased anisotropic index and an increased hardness and cutting force. The study showed that the addition of hydrocolloids increased the mechanical rigidity of the paste and provided shape fidelity. Demircan et al. [80] developed formulations with the incorporation of three different mushrooms (*Ganoderma lucidum*, *Lactarius deliciosus*, and *Pleurotus ostreatus*) to produce meat analogs using 3D printing and investigated the printability. The study showed that all the inks prepared with mushrooms revealed gel-like viscoelastic behavior and good printability. Additionally, meat analogs produced with incorporated mushrooms showed lower hardness, stiffness, springiness, and chewiness values, and increased juiciness. The authors reported that meat analogs with the incorporation of mushrooms showed enhanced nutritional value and provided the release of amino acids related to umami. Wang et al. [81] studied the textural and physicochemical properties of 3D-printed pea protein-based meat analogs. The study showed that the pea protein-based meat analogs were not capable of forming a meat-like fiber structure with 3D printing or cooking in boiling water. Also, the 3D-printed meat analog showed lower hardness and higher springiness, cohesiveness, and chewiness values compared to non-printed meat analogs. The authors reported that the 3D printing provided personalization of the food products and led to softer meat analogs. In another study, Israeli et al. [82] produced 3D-printed meat analogs with three novel proteins (canola, chickpea, and potato) and studied the relationship between the physicochemical and functional characteristics of the novel proteins. The study revealed that different proteins showed significant differences in physicochemical and functional properties. Additionally, the study showed that there was a negative correlation between gelation temperature and solubility, as well as between water absorption capacity and the charge of the protein, with high temperatures increasing solubility and the weaker negative charge of the protein increasing the water-holding capacity. The authors reported that 3D-printed meat analogs produced using canola and potato protein showed high values of hardness, chewiness, gumminess, and free water content. The authors suggested that novel proteins interacted well with other components in the meat analog and led to a strong and moist texture. Qiu et al. [83] produced 3D-printed meat analogs from wheat gluten, soy, and rice protein, and evaluated the printing performance, morphology, and textural properties. The authors reported that the samples containing 1:1:0.7 and 1:1:1 soy protein isolate:wheat gluten:rice protein showed the highest printability performance viscosity and mechanical strength, higher stability, and similar textural properties. Thus, an increased concentration of rice protein improved the suitability for printing. Additionally, considering the results of apparent appearance and microstructure, hot-air 3D printing was evaluated as a potential manufacturing method to produce meat analogs from a mixture of soy protein isolate, wheat gluten, and rice protein. Chao et al. [84] produced 3D-printed meat analogs using mung bean protein isolate, wheat gluten, and l-cystine, and evaluated the addition of l-cystine on the functionality and formation of fibrous structures in meat analogs. They reported that the addition of l-cystine of up to 0.4 wt% improved the printing properties and structural stability, which correlated with increased mechanical strength and intermolecular cross-links. Additionally, the 3D-printed meat analog that contained 0.4 wt% showed a muscle meat-like structure and enhanced textural properties, and after the cooking process, it showed a more compact and pronounced fibrous structure with a decreased beany odor and bitter taste. In the study by Cheng et al. [85], 3D-printed meat analogs were produced using soy protein isolate, wheat gluten, insoluble dietary fiber, beet red, and deionized water. They evaluated the printing performance, molecular interactions, morphology, and physicochemical, textural, and rheological properties. They reported that the addition of insoluble dietary fiber improved the hardness, chewiness, gumminess, WHC, tensile strength, and elongation at break, and enhanced the disulfide bonds, leading to improved printing properties.

Kang et al. [86] investigated the viability of 3D-printed meat analog production using insect protein (*Tenebrio molitor* larvae) with varied concentrations (5, 10, and 15 wt%) and different fractions (filtrate, supernatant, and pellet). The study showed that increased concentrations of the insect protein in the filtrate or pellet fractions provided increased extrudability and water-holding capacity, strong uniform internal structure, and a decreased deformation rate after printing. The authors reported that the protein concentration was capable of modifying the textural properties, with the highest hardness and chewiness obtained with 15 wt% insect protein. In the study by Nam et al. [87], the rheological and physicochemical properties of a 3D-printed soybean- and insect protein (*Gryllus bimaculatus*)-based meat analog were investigated. The study showed that among the *Gryllus bimaculatus* fractions (pellet, supernatant, and filtrate), the filtrate fraction can be used to produce high-calorie meat analogs, and supernatant and pellet fractions can be used to produce a meat analog appropriate for a high-protein, low-fat, and low-carbohydrate diet. Additionally, the highest recovery rate, highest protein content, lowest moisture, and an appropriate amount of fat and carbohydrates were found in the pellet fraction of the insect protein. Thus, the authors reported that enhanced nutritional content in addition to improved physicochemical and structural characteristics was found in the 3D-printed meat analogs using inks containing 15 wt% pellet.

Additionally, the review by Tibrewal et al. [98] includes production technology with different types of proteins; however, their study was excluded based on E1 (review article).

### 3.4. Main Findings of the SLR for Fish Analogs

All selected articles on fish analogs are compiled in Table 4 and divided into five categories: (i) type of analog, (ii) protein type, (iii) composition of the meat analog, (iv) tested parameters, and (v) technology used to obtain the fish analog (Table 4).

The alternative proteins used in the selected studies are shown in Figure 6.

Pulse proteins were used in 12 studies alone or with a combination of other proteins, followed by macroalgae, microalgae, plant, cereal, and fungal proteins (used in one study). It was observed that pulse proteins, which include pea, soy, red lentil, and yellow lentil proteins, were one of the principal alternative protein sources to produce fish analogs. Also, kombu, *Nannochloropsis oceanica*, and nori are algae proteins with some relevance in the published literature. Additionally, oyster mushrooms were used as a fungal protein in the developed formulations. Additionally, duckweed and brown rice protein were included in the plant and cereal proteins, respectively.

The high utilization of pulse proteins in fish analogs might be related to their promising ability to mimic fish muscle structure [114]. Additionally, it seems that the utilization of algae proteins to produce fish and seafood analogs is higher compared to meat analogs. This can be explained by the fishy flavor of algae-based proteins, which may be an undesirable sensory aspect for meat analogs [115]. The rise in the utilization of algae-based protein sources in fish analogs might be related to their well-balanced essential amino acid composition and high sustainability, mainly due to low land use [22].

According to the years for each protein, the earliest protein to produce fish and seafood analogs was pulse proteins (Figure 7).

As fish and seafood analogs are still a new topic in the food area, and pulse proteins were used widely in the food industry to produce meat analogs, it is reasonable to explore new food analogs with a protein that is more available. However, with the increased concern for sustainability and health issues such as allergenicity that can be caused by pulse proteins (i.e., soy protein), new sources of proteins are starting to be explored [116].

It was observed that conventional technology mixing was used in nine studies (69%), while the novel technology of 3D printing was used in four studies (31%). Fish analogs, such as salmon fillets, exhibit complex fibrous morphology to mimic [114]. Three-dimensional-printing technology is capable of producing solid forms, such as food analogs, through layer-by-layer printing, enabling the binding of those layers through chemical reactions or phase transitions. Thus, 3D printing facilitates the modification of rheological and textural properties [117] and seems to be a promising technology for the production of complex food analogs such as fish.

### 3.5. Overview of the Studies Selected for Fish Analogs

#### 3.5.1. Mixing

The selected articles indicate that fish analogs based on alternative proteins started to be studied in early 2022, and it is a new topic in the food science and technology field. For instance, Ran et al. [102] investigated the effect of the addition of KGM at different concentrations (3.5%, 5.0%, 6.5%, and 8.0%) on the textural and rheological properties of soy protein-based fish ball analogs. The addition of KGM at lower concentrations (3.5% and 5.0%) led to fish analogs with a loose and weaker gel structure due to the presence of more and larger pores in the gel network. On the other hand, the addition of KGM at higher concentrations (6.5% and 8%) provided denser crosslinks with the protein, a firmer gel structure, and enhanced hardness and chewiness. The authors reported that a more compact gel structure and higher values for the elastic modulus were achieved at higher concentrations of KGM (6.5% and 8%) in the formulations. Similarly, in the study by Ran and Yang [103], the microscopic and macroscopic characteristics of the fish ball model system containing a KGM–soy protein complex were studied. The study showed that the concentration of KGM directly affected the gel strength, crystallinity, strain hardening, and dynamic mechanical thermal behavior, with the increased concentrations of KGM providing stronger gels, higher melting temperatures, and an increased relative crystallinity index. The authors reported that the incorporation of KGM enhanced the rheological, functional, and textural properties of the model of soy protein-based fish analogs. Zhang et al. [104] produced scallop analogs from enzymatic gelation of pea protein–pectin mixtures and studied the microstructure, textural, and functional properties. The study showed that different pectin concentrations were able to modify the microstructure and physical properties of the scallop analogs, with lower pectin concentrations providing stronger gels (up to 0.5 wt%). The authors reported that the water-holding capacity of the scallop analogs was decreased and the pore size was increased with higher concentrations of pectin. Interestingly, the addition of 0.5 wt% pectin to the scallop analogs showed similar microstructure, textural, and physicochemical properties to real scallops. Kobata et al. [105] produced a seafood sea foie gras analog using duckweed RuBisCO protein, flaxseed oil, and β-carotene through mixing. The authors evaluated microstructure, morphology, appearance, and physicochemical, textural, and rheological properties. They reported that the formulations containing β-carotene, 40 wt% flaxseed oil, and 10% *w*/*v* duckweed RuBisCO protein showed similar color and textural properties to a real seafood product. Additionally, the sea foie gras analog showed a less fibrous structure, yet showed a similar fat droplet distribution and stronger molecular interactions in the gel network. The authors suggested that for future studies, the sea foie gras analog should be evaluated through in vitro and in vivo tests to determine the bioaccessibility and bioavailability of the proteins, carotenoids, and omega-3 fatty acids. Moreover, the authors recommended determining the sensory characteristics of the sea foie gras analogs to compare them to a real sea foie gras product. Ran et al. [106] investigated the effect of deep-frying and air-frying soy protein-based fish ball analogs on physicochemical properties and mass and heat kinetics. The study showed that the deep-frying process showed more effective moisture and heat transfer, increased oil uptake, fast texture development, and higher total color change, resulting in a darker crust color. Additionally, during frying, the soy protein-based fish ball analog showed a rough and decreased uniform structure. The authors reported a strong correlation between quality attributes and mass and heat kinetics, which provided insights for the fish analog industry. In the study by Zhang et al. [107], the digestibility of the pea protein- and pectin-based scallop analogs using an in vitro digestion model was investigated and compared with scallops. The scallop analog showed lower protein digestibility compared to scallops, and the authors suggested that the low protein digestibility might be due to the nature of the protein, including antinutritional factors that pea protein could contain such as phytates, tannins, trypsin inhibitors, and lectins, which can delay the protein hydrolysis, or the possible impact of the pectin on the hydrolysis of the proteins. The authors proposed that the utilization of proteins from different sources (e.g., cereals and legumes) would be an important factor to increase the protein digestibility in the fish analogs. In the study by Peh et al. [108], fish cake analogs were produced using brown rice and pea protein isolates, with the addition of MC, CG, and high acyl gellan gum through mixing. They evaluated the microstructure, amino acid profile, and physicochemical, textural, and rheological properties of the fish cake analogs. They reported that the fish cake analog containing CG showed an increase in heated water-holding capacity for higher concentrations of CG, while the heated oil-holding capacity was increased only with increased concentrations of the MC. They evaluated the textural properties of the fish cake analogs at two temperatures: 4 °C and 55 °C, the storage and serving temperatures, respectively. The sample containing MC showed higher hardness for all concentrations compared to other gums. Additionally, the profile of amino acids after in vitro digestion mainly constituted polar acidic amino acids for all types of gums. The authors reported that the combination of MC, CG, and high-acyl gellan gum improved the textural properties of the fish cake analogs, and the fish cake analog produced with 1% MC, 3% CG, and 1.5% GG showed the most similar hardness and springiness values to the commercial surimi-tofu fish cake (hardness: 1006 g at 4 °C, 1385 g at 55 °C; springiness: 0.32 at 4 °C, 0.33 at 55 °C).

In another study, Patil et al. [109] optimized the development of macroalgae-based (nori and kombu) fish analogs using D optimal design and fuzzy logic methods. The study showed that both the D optimal design and the fuzzy logic method provided an optimum solution for the formulation of a fish analog with 8% nori, 6% kombu, and 59.4% oyster mushroom. The study showed that a citric acid and potassium sorbate mixture as preservatives retain the antioxidant capacity and flavonoid content. The authors reported that the treatment with citric acid and potassium sorbate mixture showed no detection of bacterial, yeast, or mold count on the 30th day of the fish analog, which led to the safe consumption of these fish analogs after an air-frying process. Zhao et al. [110] investigated the effects of the incorporation of microalgae (*Nannochloropsis oceanica*) into a pea protein-based fishcake analog on physicochemical properties and digestibility. Pea protein was replaced with microalgae at different concentrations (0%, 10%, 20%, and 30%). The study showed that higher concentrations of microalgae (30%) in the formulation increased the hardness, juiciness, elasticity, and water-holding and oil-holding capacity of the fishcake analog. Additionally, the incorporation of the microalgae increased the protein digestibility compared to the fishcake analog without microalgae. However, when increasing the concentration of microalgae in the formulations, a decrease in protein digestibility was observed, with the 10% incorporation showing 74.9% and the 30% incorporation showing 70.1%, because of limited protein–enzyme interactions due to higher viscosity and the existence of indigestible compounds. The authors reported that the release of certain amino acids and fatty acids was decreased, and a more compact structure was obtained with the incorporation of the microalgae.

#### 3.5.2. 3D Printing

In the study by Lee et al. [111], soy protein-based fish analogs were produced using uniaxial-nozzle 3D printing to obtain a fibrous structure. The study showed that soy protein-based inks showed gel properties (G′ > G″); however, they showed lower dimensional stability and printability, according to the results of the thermal behavior tests. The authors reported that cutting strength was affected by the fiber thickness based on the nozzle size, number of columns, and fiber direction, and the texturization degree increased with the increased fiber thickness based on the column number or nozzle size. Magarelli et al. [112] produced a soy protein-based salmon fillet analog using 3D printing and evaluated antioxidant activity as well as total phenolic compounds such as isoflavone. The study showed that the production of a salmon fillet analog with pulse proteins is possible, and the detected isoflavone content is considered safe for daily consumption. In the study by Shi et al. [113], the textural properties and the effect of the printing parameters of 3D-printed soy protein-based fish analogs were investigated. The study showed that it is possible to produce a low-cost and protein-rich fish analog from soy protein, xanthan gum, and rice starch using 3D printing. Additionally, formulations containing 20 wt% soy protein, 3 wt% xanthan gum, and 15 wt% rice starch showed stable printability and regular and continuous filaments due to an appropriate amount of water mobility and the binding force of hydrogen protons. The authors reported that the nozzle size and porosity parameters were able to modify the textural properties of the fish analogs, with increased nozzle size and increased porosity decreasing the hardness, chewiness, and gumminess. Tay et al. [114] investigated the influence of high-pressure homogenization and post-printing TGase treatment of a 3D-printed salmon fillet analog on microstructure and physicochemical characteristics. The study showed that the sample HPH HO-LP (formulation treated with HPH and containing high oil (45 wt%)–low protein (5.5 wt%)) showed the highest values for the storage (G′) and complex modulus (G*). Also, according to the results of rheology, the same sample showed solid viscoelastic behavior and the greatest mechanical strength. Additionally, it was observed that the application of HPH increased the hydrophobicity of the proteins; thus, oil droplets were absorbed by a higher number of protein particles and led to gel formation. Furthermore, the authors aimed to mimic the texture of a salmon fillet by preparing simulants of myosepta (white fat tissue) and myomere (orange muscle tissue). They prepared the myosepta using deionized water, camelina oil, TGase, and yellow pea protein, and they prepared the myomere using deionized water, camelina oil, TGase, and red lentil protein. After the addition of myosepta and myomere to the fish fillet analogs, the authors reported that the myomere simulant affected the texture profile of the 3D-printed salmon fillet analog more than the myosepta simulant, and the addition of TGase provided a stable texture during conventional cooking.

Furthermore, Coleman et al. [115] studied the potential of microalgae as flavor agents for plant-based seafood analogs. In their study, they identified the odor and taste characteristics of each selected microalgae, and they determined two microalgae species (*Tetraselmis chui* and *Phaeodactylum tricornutum*) as potential flavoring agents for plant-based seafood analogs. However, their study was excluded in this SLR based on E5 (no final food analog production).

### 3.6. Bibliometric Analysis

#### 3.6.1. Network Visualization by Keyword for Meat Analogs

Keyword analysis was performed using the authors and indexed keywords. The keyword network for the meat alternatives is presented in Figure 8, and each node represents a keyword.

The size of a node indicates the number of published records related to the keyword, and the color of the node is related to the publication year, with the darker purple representing the keyword having appeared in the early 2010s and yellow representing the keyword having appeared recently (2023). Lines indicate the links between the keywords, and the thickness of the lines represents the likelihood of co-occurrence of keywords in the same publication. The minimum number of occurrences of keywords was two. Of 374 keywords, 73 met the threshold and a total of 73 keywords formed 3 clusters, with 631 links, and the total link strength was calculated as 787. The most frequently occurring terms were “meat analog” (17 documents), “soy protein” (13 documents), “3D printing” (11 documents), “textural properties” (10 documents), and “rheological properties” (10 documents), followed by the terms “food” (9 documents) and “extrusion” (7 documents) Additionally, it is possible to observe that alternative proteins from a fungal origin started to be used as an ingredient in the early 2010s. It seems that there is a strong connection between the keywords “meat analog” and “3D printing”, which co-occurred in seven documents (15%). Additionally, the texture of the meat analogs seems an important parameter contributing to their development, as “meat analog” and “textural properties” co-occurred in four documents. Also, “meat analog” and “rheological properties” co-occurred in five documents, “3D printing” and “rheological properties” co-occurred in five documents, and the average publication year of the term “3D printing” was calculated as 2022.45 (mid-2022). In this research, the selected studies showed that the earliest meat analogs were produced using extrusion (2003) or mixing (2011), which may show limitations in sensory (i.e., shape) or textural properties of the final product. However, the recent developments show that 3D food printing allows for control of the textural, nutritional, and sensory properties of the final product based on the selection of raw materials and the design of the formulations [118]. Thus, it appears that edible materials with complex structures, such as meat analogs due to muscle-fiber orientations, can be produced using this technology.

#### 3.6.2. Network Visualization by Author for Meat Analogs

The co-occurrence network visualization for the authors of the selected records is shown in Figure 9.

Each node represents a different author, and the node size is directly related to the number of records published by the author, with bigger nodes meaning that the author has more published records. Additionally, nodes that are closer together indicate a more relevant bibliographic connection between the authors. Based on the 46 selected records, a total of 202 authors conducted their studies in the area of meat analogs, and they formed 29 clusters with 556 links, and the total link strength was calculated as 601. Overall, 169 authors had only one publication in the list of selected records. The highest number of publications was found for the authors Kim H.W. and Park H.J., with five each, followed by Ettelaie R., Toepfl S., Chen J., and Wen Y., with three publications. Additionally, the largest set of connected authors consisted of 17 authors, including Ahmad I., Bian M., Chen J., Cheng Z., Dai T., Ding Y., He Y., Li C., Li Y., Liu C., Lyu F., McClements D. J., Qiu Y., Wu X., Xu S., Zhang C., and Zhou J. Additionally, Toepfl S. had the highest number of citations (245 citations), followed by Palanisamy M. (171 citations), Kim H.W. (128 citations), Park H.J. (128 citations), and Wen Y. (124 citations). The results suggest that Kim H.W., Park H.J., Toepfl S., Palanisamy M., and Wen Y. are highly productive researchers in the area of the development of meat analogs. Additionally, the affiliations of the authors were screened, and 51% of the authors conducted their research in universities or research organizations on the Asian continent.

#### 3.6.3. Network Visualization by Keyword for Fish Analogs

The keyword network for fish alternatives is presented in Figure 10, with the darker purple representing that the keyword appeared in early 2022 and yellow representing that the keyword appeared recently (2024).

The lines indicate the links between the keywords, and the thickness of the lines represents the likelihood of co-occurrence of keywords in the same publication. The minimum number of occurrences of keywords was two. Of 153 keywords, 24 met the threshold. It was found that a number of authors utilized variants of keywords. The most frequently occurring terms were “textural properties” (6 documents), “rheological properties” (6 documents), “3d printing” (4 documents), and “protein” (4 documents). A total of 24 keywords formed 3 clusters, with 136 links, and the total link strength was calculated as 166. It is possible to observe that KGM was used in early 2022 for fish analogs. It seems that there is a connection between keywords “rheological properties” and “textural properties”, with co-occurrence in three documents. Additionally, the textural and rheological properties of the fish analogs seem to be important parameters contributing to their development, as both “plant-based seafood analog”–“textural properties” and “plant-based seafood analog”–“rheological properties” co-occurred in two documents. Additionally, the average publication year of the term “3d printing” was calculated as 2023; thus, it appears that the novel technology 3D printing started to be used in the development of fish analogs recently.

#### 3.6.4. Network Visualization by Author for Fish Analogs

The co-occurrence network visualization of the authors for the selected records is shown in Figure 11.

Based on the 13 selected records, a total of 50 authors conducted their studies in the area of fish analogs. They formed 9 clusters, with 115 links, and the total link strength was calculated as 126. Overall, 44 authors had only one publication in the list of selected records. The highest number of publications was found for the authors Ran X. L., Yang H. S., Kobata K. McClements D. J., and Zhang Z. Y., with three publications. Additionally, the largest set of connected authors consisted of nine authors, including Kobata K., Kos D., Lu J. K., McClements D. J., Pham H., Qin D. K., Rao J. J., Tan Y. B., and Zhang Z. Y. Also, Ran X. L. and Yang H. S. had the highest number of citations (122 citations). The results indicate that Yang H. and Ran X. are highly productive researchers in the field of fish analog development. Similarly, the affiliations of the authors showed that 67% of the authors conducted their research in universities or research organizations on the Asian continent.

## 4. Discussion

To answer RQ1, a total of 46 articles were found using the selected keywords and Boolean operators for meat analogs. The oldest publication year was found to be 2003, and the latest was 2024. The publication trends of meat analogs showed that the number of publications was increasing day by day, and the year of the peak number of publications was 2023, with 18 publications. However, it should be kept in mind that this research includes only studies available by 31/05/2024. Therefore, based on the exponential increase, the number of publications in 2024 and in the coming years ahead is expected to increase. For fish analogs, a total of 13 records were found between 2022 and 2024 using the selected keywords and Boolean operators for this research. It was observed that the number of publications increased exponentially from three to eight within one year (2022 to 2023), and two publications were found for the year 2024. Additionally, there was a significant difference in the number of studies found for meat and fish analogs, with the meat analogs constituting 46 studies and fish analogs constituting 13 for similar selected keywords. This difference may be related to the changes in consumers’ perceptions, because the consumers can avoid or decrease their consumption of meat products due to related health issues caused by high saturated fat content [5]. Additionally, as fish products show a balanced lipid profile, with a similar protein content to meat products [7], consumers can increase their consumption of fish products. However, lately, concern about the environmental impact of and animal welfare regarding meat and fish products has increased [4,9], and a search for alternatives to fish products to decrease consumption and decrease the negative environmental impact has begun. The results showed that there was a higher number of published records on the production of meat analogs using alternative proteins and novel technologies than for fish analogs. Therefore, the production of fish analogs using alternative proteins and novel technologies is considered a recent area in food sciences compared to meat analogs.

RQ2 was focused on the production technologies used in meat and fish analogs. For meat analogs, it was observed that the main production technology was extrusion, followed by the novel technology of 3D printing and mixing. For fish analogs, the main production technologies were identified as mixing, followed by 3D printing. Mixing is a conventional technology, and it can be manual or involve the utilization of appropriate equipment such as a food blender at high or low speeds. It uses the top-down approach during the production of solid forms, such as meat and fish analogs, and provides an anisotropic, fibrous structure by shearing the mixture of proteins and/or polysaccharides. It was reported that a top-down approach provides less hierarchical fibrousness in meat analogs than in meat [33]. Additionally, during the production of food analogs with mixing, fiber formation can be initiated with the addition of casein to the emulsion to entrap the anisotropic structure [119]. However, despite providing enhanced fibrous structure, this processing method shows some limitations in terms of resources that are used, as production of the final product may require several steps and raw materials (e.g., structuring agents) [120]. Likewise, extrusion technology uses the top-down approach to produce food analogs, and the produced food analog might show undesirable rheological, textural, and physicochemical properties. However, the extrusion type (e.g., high or low moisture) and parameters (e.g., screw rotation speed) are capable of modifying the rheological, textural, and physicochemical properties of the extruded materials, which enables the production of food analogs such as meat analogs with similar characteristics to meat [121]. Nevertheless, currently, one of the main challenges of extrusion technology is related to high energy usage [120]. Similarly, the novel technology of 3D printing uses the top-down approach during the production of solid materials, and meat analogs produced with 3D printing might show fibrousness characteristics that are inferior to those of meats [33]. However, 3D printing enables the personalization of food analogs, where the nutritional composition, shape, and size may be modified depending on the computational design [100]. Also, printing parameters such as nozzle-moving speed, extrusion rate, nozzle diameter, layers, nozzle height, and temperature affect the printability of the food; thus, changing these parameters allows for the textural and rheological properties to be modified [122]. Despite the increased ability to control the final textural and rheological properties, the implementation of 3D printing on an industrial scale has limitations due to printing speed and the high initial investment. Moreover, currently, consumers express doubts about the quality and nutritional composition of 3D-printed food products, and these doubts lead them to be more resistant or skeptical about including 3D-printed food products in their diet [99]. However, the utilization of 3D printing in the production of food analogs is expected to increase in the upcoming years to overcome those challenges, as important quality parameters can be modified and optimized by changing 3D-processing parameters and printing materials.

The main types of alternative proteins that have been used to produce meat and fish analogs were discussed to answer RQ3. For meat analogs, the most widely used alternative proteins were from pulses, followed by cereals, fungi, microalgae, and proteins from other sources, such as oilseed, flowering plants, tubers, microbial single-cell protein, mushrooms, and insects. The records included in this study indicated that mycoprotein was the earliest protein source used to produce meat analogs, as the first utilization was in 2003, followed by pulse and cereal proteins in 2011. Additionally, proteins from microalgae and insects were started to be used to produce meat analogs in 2018. Furthermore, other proteins, such as canola, duckweed, and potato, were used in meat analog production starting in 2023. Similarly, for fish analogs, pulse proteins were the most widely used alternative protein sources, followed by algae and fungal proteins. According to the years for each protein, the earliest protein used to produce fish and seafood analogs was pulses (2022), followed by fungi, plants, macroalgae (2023), microalgae, and cereals (2024).

Pulse proteins, such as lentil, cowpea, chickpea, faba bean, green gram, horse gram, lupin, mung bean, pea, and soy, showed balanced nutritional properties due to a high amount of lysine, leucine, aspartic acid, glutamic acid, and arginine content [88]. Additionally, during their production, they show high sustainability, which is an important aspect considering the increased impact of food production on global warming [16]. Also, their functional, physicochemical, rheological, and textural properties can be modified with several processing methods, such as biological (e.g., enzymes and fermentation), chemical (e.g., acylation, deamidation, glycosylation, and phosphorylation), and physical (e.g., extrusion, cold plasma, heat, HPH, and ultrasound treatment), which enables their behavior in the food matrix to be improved [123]. Similar to pulse proteins, cereal proteins such as wheat gluten and rice contain dietary fiber, phenolic compounds, and unsaturated fatty acids. Also, their high composition of bioactive peptides decreases the risk of chronic diseases, and they should be included in the diet to prevent obesity, CVDs, diabetes, and high cholesterol [89]. Additionally, as cereal proteins contain high amounts of glutamine and asparagine, they can be modified to increase their functional properties and decrease the allergenicity using chemical methods such as deamidation [124].

Mycoprotein is one of the fungus-based proteins most used as an animal protein substitute due to its high content of protein and low energy profile. It is mainly produced from the fermentation of filamentous fungus, such as *Fusarium venenatum* [125]. It is believed that mycoprotein was discovered in the late 1960s, and since then, it has been used in the human nutrition [126]. Mycoprotein shows high nutritional value of dietary fiber, minerals, and vitamins in addition to proteins, and contains a low content of fat, which mainly consists of polyunsaturated fatty acids such as ω-6 and ω-3 fatty acids [20]. However, similar to other proteins, mycoprotein also might show allergenicity [126], which might be decreased with chemical, physical, or biological modifications [124].

It is believed that the consumption and utilization of algae starts at the era of early humans due to their high nutritional and health properties [127]. Algae production shows high sustainability, and algae proteins, depending on the species, contain an increased number of essential amino acids that humans cannot synthetize. Thus, its consumption to replace animal proteins is increased to minimize the negative environmental impact and to increase nutritional quality in the human diet. Depending on the bioactive compounds they contain, *Arthrospira* spp. show different colors. Phycocyanin is responsible for the blue, chlorophyll is responsible for the green, and carotenoids are responsible for the yellow-red color. The quantity of *Arthrospira* spp. incorporated in food products should be well studied, because high quantities can cause an undesired increased intensity of colors such as blue, green, yellow, or red, thus impacting the overall acceptability of the final product. Color is an important parameter for the liking of food by consumers; therefore, the developed product might not be evaluated with higher scores of liking depending on the final color [128]. However, its potential to be used in the control of type 2 diabetes [129] and to decrease the risk of neurological and neurodegenerative diseases [130] makes them an interesting candidate as a functional ingredient for newly developed food products. Moreover, oilseed proteins such as canola, rapeseed, and sunflower seed have recently been used to produce food analogs due to their high-quality composition. They normally show a protein content of up to 25%, especially sunflower seed protein, which may show a content of up to 50% after oil removal. After modification through fermentation or enzymatic methods, some studies have shown that these proteins can be incorporated into food products [131]. For instance, Pöri et al. [132] showed that, through lactic acid fermentation with a neutral pH shift in sunflower seed protein, it was possible to incorporate sunflower seed protein into meat analogs, and this incorporation enhanced the meaty flavor of the meat analogs. Additionally, they found that incorporation of sunflower seed into meat analog matrix provided increased hardness, similar to a meat analog produced with canola protein [82]. Although oilseed proteins currently show several limitations in the removal of phenolics and oil due to their high cost and the sensory and textural properties, the utilization of oilseed proteins in food analogs is expected to increase in the upcoming years [131].

Insects show sustainable production considering the utilization of land and water, as well as their lower contribution to deforestation and the production of greenhouse gases [133]. The consumption of mealworm (*Tenebrio molitor*) has recently been approved by the European Commission [134]. *Tenebrio molitor* shows high nutritional value and contains all the essential amino acids; polyunsaturated fatty acids such as ω-6 and ω-3 fatty acids; vitamins such as vitamin A, C, E, and niacin; and minerals such as potassium, calcium, iron, magnesium, and zinc [135]. However, the incorporation of insect protein into food matrices shows challenges such as low consumer acceptance mainly due to food neophobia, lack of optimized and cost-effective processing and extraction methodologies, environmental impact due to extended processing steps, allergenicity, and lack of new regulations and legislations that ensure their safety for human consumption [136]. For the development of meat and fish analogs, the selection of an alternative protein source should be well studied, as besides the nutritional profile, each alternative protein source shows different physicochemical and sensory characteristics, which may modify the characteristics of the final product.

Network visualization by keyword for meat analogs shows that rheological properties are important characteristics for designing meat analogs. They provide information about the deformation of the food under a determined force as a function of time at a molecular level. Additionally, the rheological behavior of the food product influences the texture–taste interactions; thus, it is associated directly with texture, taste, and mouthfeel [137]. Moreover, there is a strong relationship between meat analogs and their textural properties. Likewise, the textural properties of meat analogs are one of the quality attributes, and it may affect consumers’ overall liking [138]. Fortunately, different production technologies and ingredients are capable to modify the textural properties, which allows meat analogs with the desired and adequate textural properties to be obtained [34]. Additionally, “3D printing” and “meat analogs” were mentioned together in more than five papers, and the average year of publications for 3D printing was calculated as 2022.45 (mid 2022), which indicates that 3D printing gained more attention for producing meat analogs starting, on average, in mid-2022. This suggests that there is a growing interest in investigating the feasibility of 3D printing in food production, as it is a time-saving novel technology that also enables personalized food production in terms of shape, color, nutritional quality, and rheological and textural properties [139].

The network visualization by author for meat analogs shows that there was an important exchange of knowledge between the authors Kim. H. W. and Park H.J. based on the thickness of the links. These results indicate that the research on meat analogs with alternative proteins is increasing, considering the total number of publications and citations. Thus, it can be said that among the 202 authors, H.W., Park H.J., Toepfl S., Palanisamy M., and Wen Y. were the leading authors for the area of meat analogs.

Network visualization by keyword for fish analogs shows that the keywords “textural properties” and “rheological properties” were mentioned together with the keyword “plant-based seafood analog” in two records each. Recently, it was reported that commercial fish analogs showed lower protein content compared to conventional fish products, and besides the protein content, fish analogs should be fortified with micronutrients such as ω-3 fatty acids and vitamins A, B, and D [140]. Additionally, an important relationship between fish analogs and their textural properties was detected. Textural properties of fish analogs are considered a quality attribute that influences the overall liking by consumers, and it can be modified with different processing technologies and chemical composition [10]. Another important relationship was detected between fish analogs and their rheological properties. The muscle structure of conventional fish products shows a hierarchical structure that provides specific properties of texture, viscoelasticity, and mouthfeel. Thus, mimicking the fish muscle turns into a more complex process; however, novel technologies and types of ingredients in the formulation of fish analogs may provide adequate and desired viscoelastic properties in the final product [141]. Additionally, the average publication year of the term “3D printing” was calculated as 2023, which indicates that the viability of the novel technology to produce fish analogs is gaining more interest. Three-dimensional printing, in addition to other important benefits that were discussed earlier, allows the textural and structural properties of the final product to be modified through printing parameters and the nutritional composition of the ink [10].

Network visualization by author for fish analogs showed that the authors that published the most records were Ran X. L., Yang H. S., Kobata K., McClements D. J., and Zhang Z. Y. (3 publications). Additionally, the authors Ran X. L. and Yang H. S. were cited 122 times by other researchers. It seems that the most information exchange occurred between the authors Yang H. and Ran X. considering the thickness of the line between the two authors. Additionally, an increasing interest was detected for fish analogs with alternative proteins, with the authors Yang H. and Ran X. seeming to be highly productive and leading researchers.

Moreover, the affiliations of the authors indicated that universities or research organizations on the continent of Asia conducted more research to develop meat and fish analogs compared to Europe or North America. The higher number of studies in Asia may be related to Asia being the largest market for alternative proteins, and thus the area with the largest consumer demand [142].

Although the studies on meat and fish analogs are increasing, there are still some limitations regarding their high cost, sensory quality, processing steps, nutritional properties, and safe consumption. For instance, sensory properties of food analogs such as texture, appearance, and flavor are believed to be hard to mimic due to specific molecular interactions and physicochemical properties in meat and fish products. Additionally, high expectations of consumers can decrease the acceptability of meat and fish analogs due to dissimilarities in sensory properties [143]. Moreover, most of the food analogs are considered to be ultra-processed food due to increased processing steps or the inclusion of several different ingredients in their formulation. Thus, these products can be perceived by consumers as “unhealthy products”, as increased consumption of ultra-processed foods may lead to undesirable health problems, such as obesity and type 2 diabetes [144]. However, it is also reported that the design of the formulation of the ingredients of these products is important, because their health perception can be improved by including lower levels of salt and sugar and increasing the levels of protein, vitamins, and minerals in their composition [145]. Furthermore, meat and fish analogs may cause health problems due to the existence of endospore bacteria (e.g., *Bacillus* spp. and *Clostridium* spp.) in the final product, which can be caused by re-contamination during the extrusion process [146,147]. Additionally, it was reported that meat and fish analogs also may show undesirable gut functioning due to anti-nutrient content (e.g., protease or phytic acid) of the raw material, or cause immunological and biological impacts due to the presence of possible allergens (e.g., Gly m 3 or Gly m 4). Though these challenges can be overcome using technologies such as high pressure, pulsed light, and ultrasounds, further work is still needed to determine more detailed consequences of these processes and their possible impact on human health [148].

In the present study, we provide insight into the development of meat and fish analogs using alternative proteins and different technologies by SLR and bibliometric analysis. However, while developing this study, there were some challenges that should be addressed. As the included studies were records published until May 2024 from the databases Scopus and Web of Science, this study might show a lack of newly published records (especially after May 2024). Additionally, further studies on meat and fish analogs based on alternative proteins should focus on critical analysis of current production technologies that are used in commercially available products, mainly for their advantages and limitations. Moreover, future studies should also focus on the evaluation of sensory and digestion properties to fill the gap in the present knowledge.

## 5. Conclusions

In this study, the scope of research on meat and fish analogs based on alternative proteins produced with conventional and novel technologies to date has been shown. The research on meat and fish analogs is increasing, but to date, most studies have focused on meat analogs based on alternative proteins. This might be due to related health issues and sustainability awareness, as the environmental impact of meat and fish products is gaining attention lately because of global warming and other environmental threats. It was also found that for meat analogs, extrusion, followed by the novel technology of 3D printing and mixing, are the main production technologies. For fish analogs, the main production technologies identified were mixing, followed by 3D printing. It seems that the utilization of the novel technology of 3D printing is increasing.

Additionally, the most widely used alternative proteins to produce meat analogs were pulses, followed by cereals, fungi, algae, insects, and proteins from other sources such as oilseed, flowering plants, tubers, and microbial single-cell proteins. Similarly, for fish analogs, pulse proteins were the most widely used alternative protein sources, followed by algae and fungal proteins. It seems that the utilization of new alternative protein sources such as microalgae is increasing. However, the quantity and the source of an alternative protein should be selected considering the physicochemical and sensory characteristics of the alternative protein, as these can influence the process of incorporating the protein into the food matrix and affect the characteristics of the final food product.

The results of the keyword analysis for the meat and fish analogs suggest that protein is an important nutritional component. Its type and the quantities that are going to be included in the analog should be well studied when designing formulations. Also, the production process and technologies that are employed should consider the desired textural parameters for the meat analogs.

## Figures and Tables

**Figure 1 foods-14-00498-f001:**
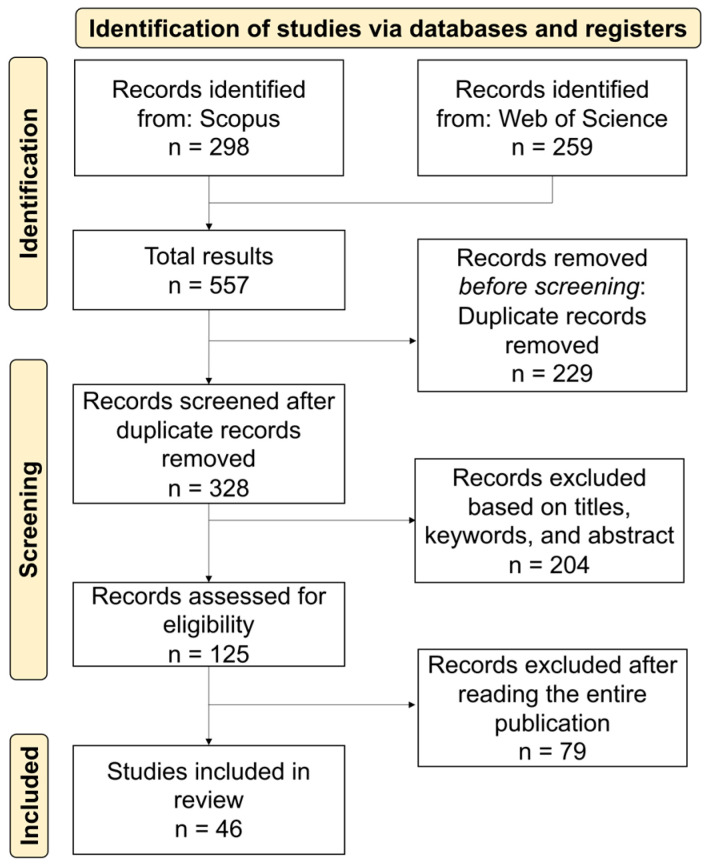
Flowchart of the identification, screening, and selection process for the SLR based on PRISMA guidelines for meat analogs.

**Figure 2 foods-14-00498-f002:**
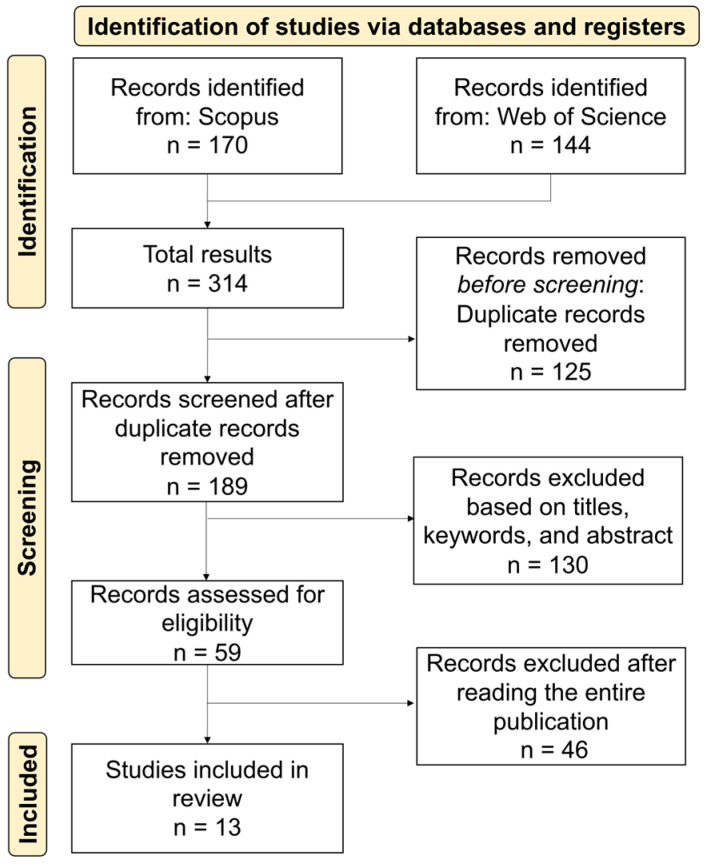
Flowchart of the identification, screening, and selection process for the SLR based on PRISMA guidelines for fish analogs.

**Figure 3 foods-14-00498-f003:**
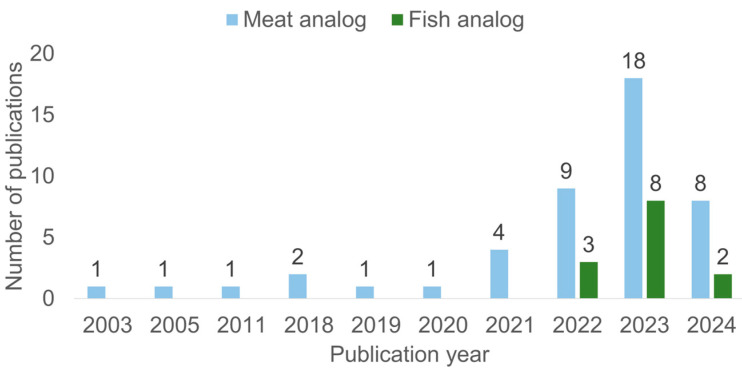
Yearly publication trends of the 46 records between 2003 and May 2024 for meat analogs (blue) and yearly publication trends of the 13 records between 2022 and May 2024 for fish analogs (green), retrieved from the Scopus and Web of Science databases.

**Figure 4 foods-14-00498-f004:**
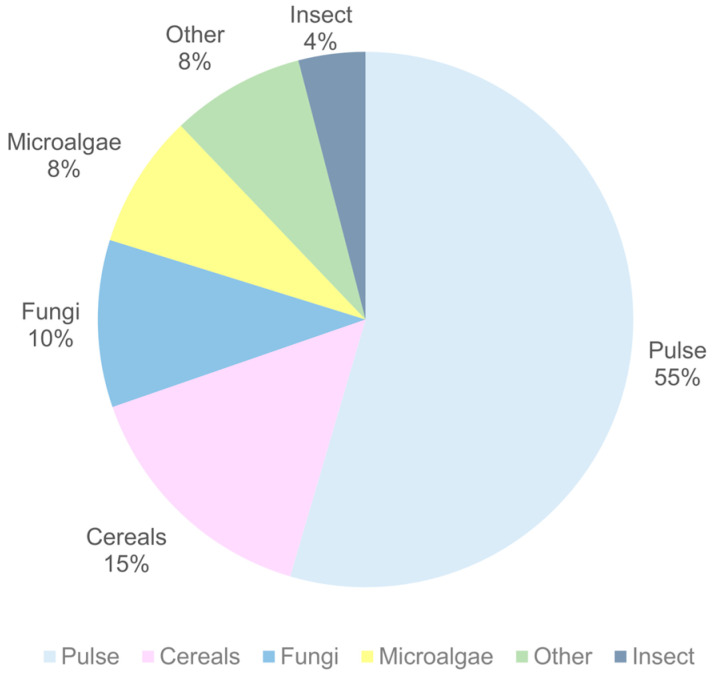
Categories of the alternative proteins that were used in the 46 selected articles on meat analogs.

**Figure 5 foods-14-00498-f005:**
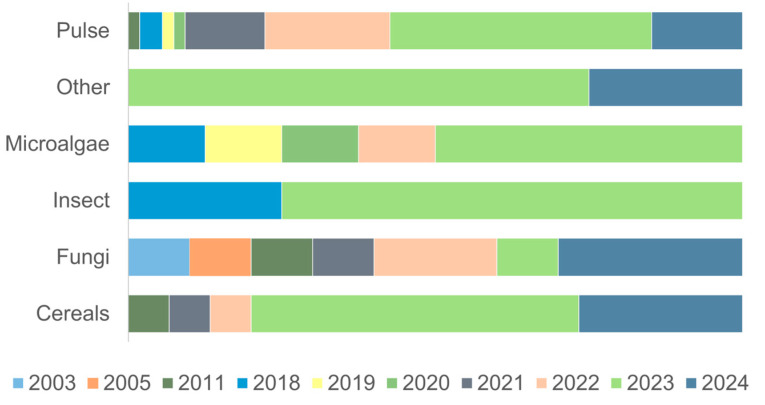
Categories of the alternative proteins that were used in the 46 selected articles on meat analogs, categorized by year.

**Figure 6 foods-14-00498-f006:**
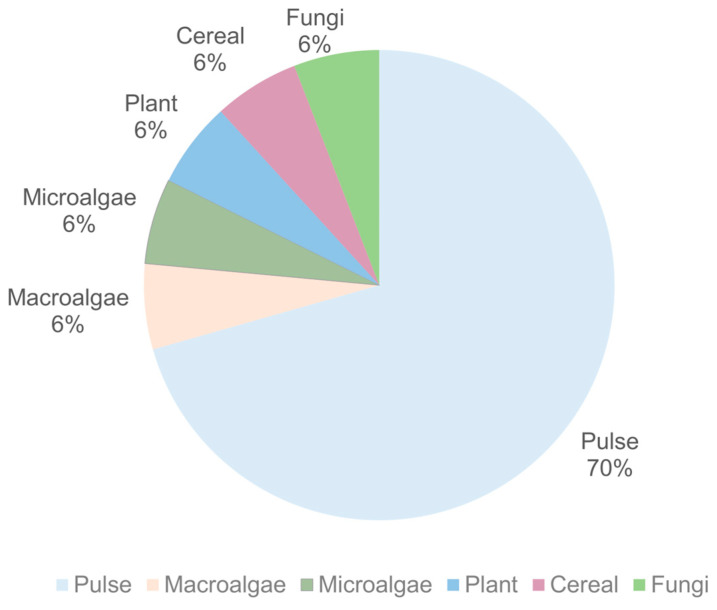
Categories of the alternative proteins that were used in the 13 selected articles on fish analogs.

**Figure 7 foods-14-00498-f007:**
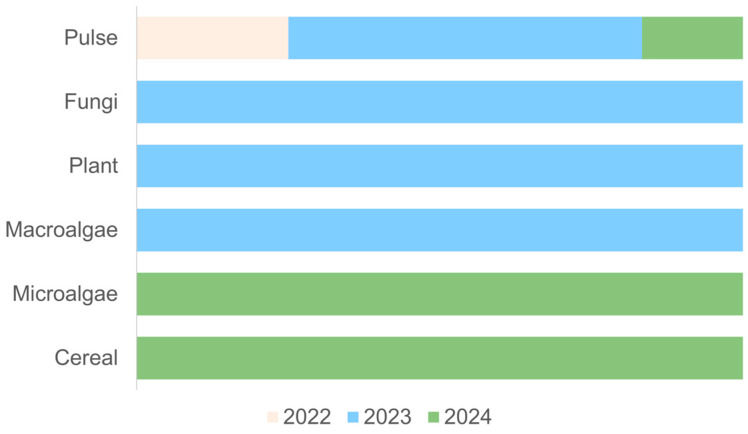
Categories of the alternative proteins that were used in the 13 selected articles on fish analogs categorized by year.

**Figure 8 foods-14-00498-f008:**
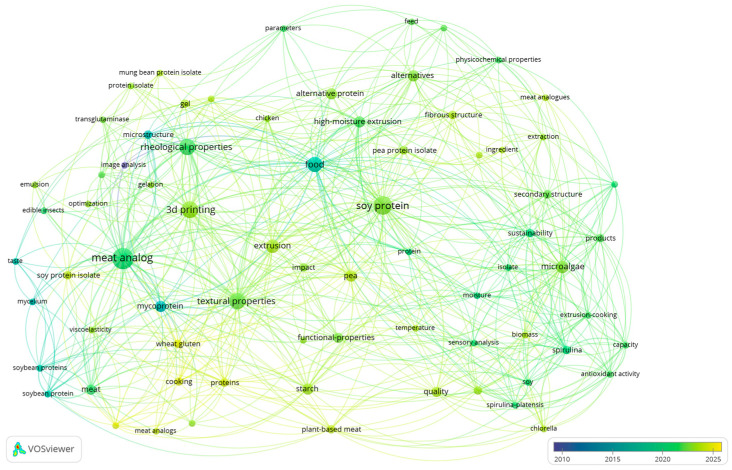
Network map of co-occurrence analysis of authors and indexed keywords using the full counting method of the selected publications for meat analogs. The analysis for the normalization method was LinLog/Modularity, the weight was the occurrences, and the score was the average publication year.

**Figure 9 foods-14-00498-f009:**
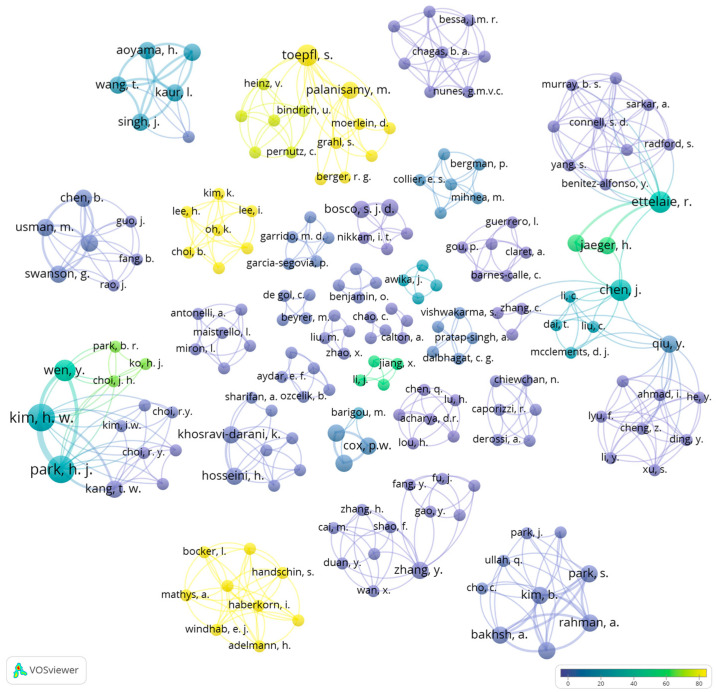
Network map of co-authorship analysis of authors for meat analogs, using the full counting method, and counting up to 25 authors per document. The analysis for the normalization method was LinLog/Modularity. The weight was the documents, and the score was the average number of citations.

**Figure 10 foods-14-00498-f010:**
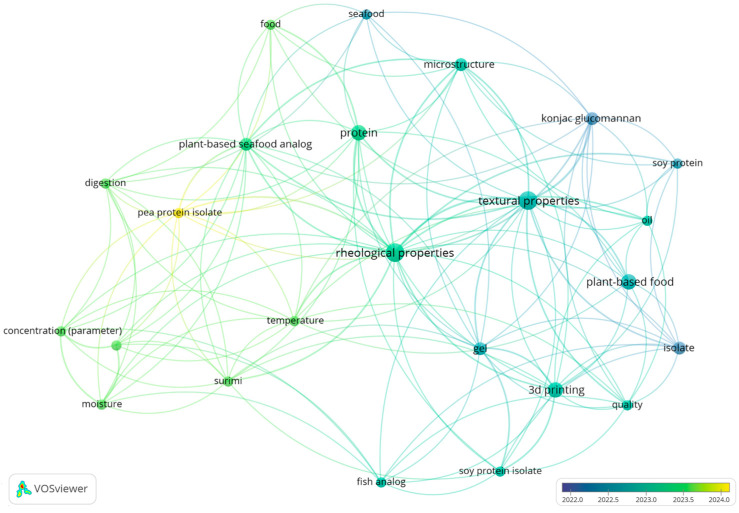
Network map of co-authorship analysis of authors for fish analogs, using the full counting method, and counting up to 25 authors per document. The analysis for the normalization method was LinLog/Modularity. The weight was the documents, and the score was the average number of citations.

**Figure 11 foods-14-00498-f011:**
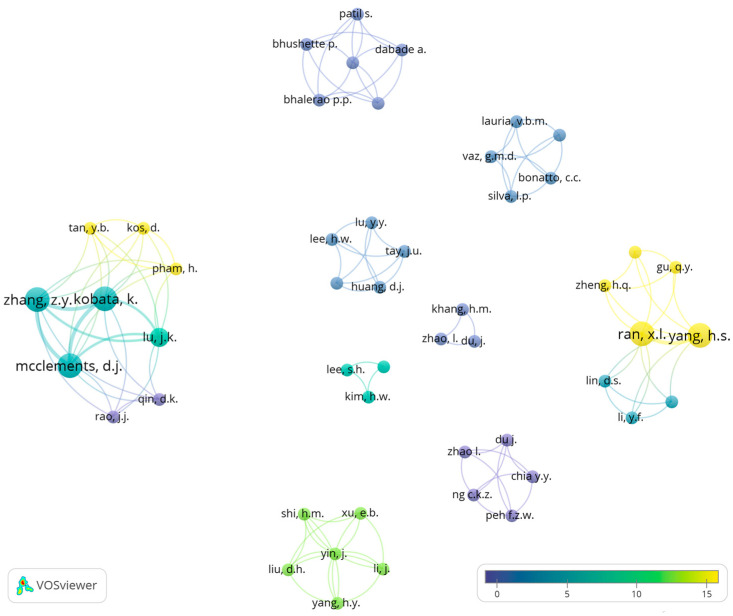
Network map of co-occurrence analysis of authors and indexed keywords using the full counting method of the selected publications for fish analogs. The analysis for the normalization method was LinLog/modularity, the weight was the occurrences, and score was the average publication year.

**Table 1 foods-14-00498-t001:** Search strings used in Scopus.

	Scopus
Meat analog	(TITLE-ABS-KEY (“meat alternative*” OR “meat analogue*” OR “meat analog*” OR “meat substitute*” OR “meat imitation*” OR “meat mimic*” OR “meat replace” OR “vegan meat” OR “faux meat” OR “mock meat”) AND TITLE-ABS-KEY (“alternative protein*” OR “pulse protein*” OR “mycoprotein” OR “microalgae” OR “3d print*”) AND NOT TITLE-ABS-KEY (“pet food” OR “animal feed”)) AND (LIMIT-TO (LANGUAGE, “English”))
Fish analog	TITLE-ABS-KEY (“fish alternative*” OR “fish analogue*” OR “fish analog*” OR “fish substitute*” OR “fish imitation*” OR “fish mimic*” OR “fish replace” OR “vegan fish” OR “faux fish” OR “mock fish” OR “seafood alternative*” OR “seafood analogue*” OR “seafood analog*” OR “seafood substitute*” OR “seafood imitation*” OR “seafood mimic*” OR “seafood replace” OR “vegan seafood” OR “faux seafood” OR “mock seafood” OR “salmon mimic*” OR “salmon fillet mimic*”) AND NOT TITLE-ABS-KEY (“pet food” OR “animal feed”) AND TITLE-ABS-KEY (“alternative protein*” OR “pulse protein*” OR “mycoprotein” OR “microalgae”) AND (LIMIT-TO (LANGUAGE, “English”))

* Is used as wildcard due to variations in language and in order to find plural forms of the words.

**Table 2 foods-14-00498-t002:** Search strings used in Web of Science.

	Web of Science
Meat analog	((TS = (“meat alternative*” OR “meat analogue*” OR “meat analog*” OR “meat substitute*” OR “meat imitation*” OR “meat mimic*” OR “meat replace” OR “vegan meat” OR “faux meat” OR “mock meat”)) AND TS = (“alternative protein*” OR “pulse protein*” OR “mycoprotein” OR “microalgae” OR “3D print*”)) NOT TS = (“pet food” OR “animal feed”) and English (Languages)
Fish analog	TS = (“fish alternative*” OR “fish analogue*” OR “fish analog*” OR “fish substitute*” OR “fish imitation*” OR “fish mimic*” OR “fish replace” OR “vegan fish” OR “faux fish” OR “mock fish” OR “seafood alternative*” OR “seafood analogue*” OR “seafood analog*” OR “seafood substitute*” OR “seafood imitation*” OR “seafood mimic*” OR “seafood replace” OR “vegan seafood” OR “faux seafood” OR “mock seafood” OR “salmon mimic*” OR “salmon fillet mimic*” AND “alternative protein*” AND “pulse protein*” AND “mycoprotein” AND “microalgae” NOT “pet food” NOT “animal feed”) and English (Languages)

* Is used as wildcard due to variations in language and in order to find plural forms of the words.

**Table 3 foods-14-00498-t003:** Findings from 46 selected articles related to meat analogs.

Type of Analog	Protein Type	Composition	Tested Parameters	Technology	Reference
High-moisture meat analog (HMMA)	Soy protein, pea protein, lentil protein, faba bean protein, gluten	Soy concentrate, soy isolate, pea isolate, pea protein, lentil protein, faba bean protein, wheat gluten, canola oil	Color, moisture content, specific density, water absorption index, water solubility index, texture profile analysis (TPA), effects of different cooling and rehydration methods	Extrusion	[42]
HMMA	Pea protein	Pea protein isolate, k-carrageenan (kc), curdlan, potato starch	Textural properties, microstructure, macrostructure, color evaluation, Fourier-transform infrared spectroscopy (FTIR), sensory evaluation	Extrusion	[43]
HMMA	Lentil protein, pea protein, gluten	Pea protein, germinated pea protein, lentil protein, germinated lentil protein, wheat gluten, salt, water, canola oil	Volatile compounds, effect of temperature and screw speed of extruder on flavor attributes, chemometric analysis, gas chromatography–mass spectrometry, effect of germination on the volatiles of pulse ingredients, effect of extrusion process on the volatiles, gas chromatography–olfactory, effect of germination on the odor profile of pulse ingredients, effect of extrusion process on the odor profile, sensory evaluation	Extrusion	[44]
HMMA	Pea protein isolate	Pea protein isolate, water	Physicochemical characterization, effect of high-moisture extrusion process on physicochemical properties, texture-related sensory evaluation	Extrusion	[45]
HMMA	Lentil protein, pea protein, gluten	Pea protein, germinated pea protein, lentil protein, germinated lentil protein, wheat gluten, salt, water, canola oil	Proximate composition, foaming capability, foaming stability, water-binding capacity, oil-binding capacity, thermal properties, protein profile, effect of extrusion process on texture, basic texture profile, morphology	Extrusion	[46]
Low-moisture meat analog	Soy protein, rapeseed protein, wheat gluten	Soy protein, rapeseed protein, wheat gluten, wheat starch, water	Specific mechanical energy, macroscopic morphology analysis, surface color analysis, expansion characteristics analysis, texture property analysis, hydration characteristics, rehydration kinetic model, micromorphology analysis, protein structure analysis, degree of gelatinization, thermal characteristics, crystal structure of the starch, protein solubility, iodine-binding analysis, residence time distribution analysis	Extrusion	[47]
Meat analog	Microalgae (*Arthrospira* spp.), soy protein	Microalgae (*Arthrospira* spp.), soy protein, water	Sensory evaluation, TPA, shear force with Meullenet–Owens razor shear	Extrusion	[48]
Meat analog	Microalgae (*Arthrospira* spp.), lupin protein mixture	Microalgae (*Arthrospira* spp.), lupin protein isolate, lupin protein concentrate, iota carrageenan, water	Cutting force, cooking yield, expressible moisture, total phenolic concentration, total flavonoid concentration, determination of Trolox equivalent antioxidant capacity, in vitro protein digestibility, FTIR	Extrusion	[49]
Meat analog	Microalgae (*Auxenochlorella protothecoides*), soy protein concentrate	*Auxenochlorella protothecoides*, soy protein concentrate, water	Cutting strength, texture profile, specific mechanical and thermal energy inputs, vitamin analysis	Extrusion	[50]
Meat analog	Microalgae (*Haematococcus pluvialis*), pea protein	*Haematococcus pluvialis* residue (HPR), pea protein, water	Textural properties, color, scanning electron microscopy (SEM), low-field nuclear magnetic resonance measurements, (water distribution), rheological measurements, FTIR	Extrusion	[51]
Meat analog	Microalgae (yellow *Chlorella vulgaris*), pea protein	Yellow *Chlorella vulgaris*, pea protein isolate, water	Moisture content, crude nitrogen content, inner texture evaluation, TPA, anisotropy index	Extrusion	[52]
Meat analog	*Haematococcus pluvialis*, soy protein, gluten	*Haematococcus pluvialis*, soy protein, gluten, complex phosphate, water	Appearance, color, texture analysis, content and proportion of volatile compound analysis, microstructure, rheological analysis, analysis of color difference value, sensory evaluation	Extrusion	[53]
Sausage	Mycoprotein	Mycoprotein, water	Microscopy and image analysis	Extrusion	[54]
Meat analog	Mycoprotein	Mycoprotein, water	Microscopy and image analysis	Extrusion	[55]
Meat analog	Pea protein, mycelium	Pea protein isolate, mycelium (*Pleurotus eryngii*), water	Physicochemical properties, color characteristics, water solubility index, water absorption capacity (WAC), oil absorption capacity (OAC), protein solubility index, rehydration properties, water-holding capacity (WHC), volumetric expansion ratio, microstructure, secondary structure	Extrusion	[56]
HMMA	Mycoprotein (*Penicillium limosum*), pea protein isolate	Mycoprotein (*Penicillium limosum*), pea protein isolate, water	Surface appearance, microstructure, WAC, OAC, protein solubility, TPA, particle size distribution, FTIR, in vitro protein digestibility	Extrusion	[57]
Meat analog	Insect (*Alphitobius diaperinus*) protein concentrate, soy protein	*Alphitobius diaperinus* protein concentrate, soy protein concentrate, water	Water content, crude protein content, protein solubility, texture analysis, SEM	Extrusion	[58]
Meatball	Green gram protein (GG), horse gram protein (HG, CPP), cowpea protein (CP)	GG, HG, CP, spice mix, meat masala, salt, corn flour, black pepper, ginger garlic paste, chopped onions, coriander leaves, baking soda, potato starch, beet root	WAC, OAC, protein solubility, foaming properties, emulsifying properties, gelling capacity, sensory evaluation, proximate composition	Mixing	[59]
Patty	Textured vegetable protein, soy protein, gluten	Textured vegetable protein, shiitake mushrooms, soy protein isolate, wheat protein isolate, tapioca starch, fats, salt, seasoning, methylcellulose (MC), garlic powder, molasses, ice, natural pigments (anthocyanin, fe-chlorophyll, dilute red, dilute red 2, red color CG2, paprika, monascus color no. 30, red rr, purple grape, cherry red, monascus color 100, red cabbage liquid, red cabbage 100, af beet red 30, grape skin color, red color pb, myoglobin)	Visible appearance, color measurements, moisture, crude fat, ash content, TPA, absorbance level, analysis of 2-diphenyl-1-picrylhydrazyl radical scavenging activity	Mixing	[60]
Meatball	GG, HG, CP	GG, HG, CP, spice mix, meat masala, salt, corn flour, black pepper, ginger garlic paste, chopped onions, coriander leaves, baking soda, potato starch, beet root	Color characteristics, morphological analysis, gas chromatography–mass spectrometry analysis, thermal properties, structural characterization, TPA, sensory analysis	Mixing	[61]
Hamburger	Pea protein	Pea protein, lucerne powder, spinach powder, chlorella powder, MC, spice mixture, olive oil, salt, pepper, garlic powder, brewer’s yeast	Water content, water solubility index, water absorption index, hygroscopicity, instrumental color, WAC, cooking loss, TPA, sensory analysis	Mixing	[62]
Patty	Duckweed protein, microalgae (*Arthrospira* spp.), yellow chlorella, textured vegetable protein, soy protein, gluten	Duckweed protein, microalgae (*Arthrospira* spp.), yellow chlorella, extruded vegetable protein, soy protein isolate, wheat protein isolate, MC, shiitake mushrooms, tapioca starch, smoked flavor, emulsion (lecithin + oleogels), salt, seasoning, coconut oil, color, garlic	Visible appearance, color measurement, proximate composition, TPA, sensory evaluation, micronutrient analysis, bicinchoninic acid protein assay, 2-diphenyl-1-picrylhydrazyl radical scavenging activity	Mixing	[63]
Hamburger	Microalgae (*Arthrospira* spp.), lentil protein	Cashew fiber, lentils, microalgae (*Arthrospira* spp.), corn starch onion powder, soy oil, salt, granulated garlic, dehydrated parsley, black pepper	Moisture, protein, lipids, ash, sensory evaluation	Mixing	[64]
Patty	Mycelium, soy protein, gluten	Soybean, water, wheat gluten, corn starch, sodium chloride, freeze-dried mushroom mycelium	TPA, SEM, moisture content, sensory analysis	Mixing	[65]
Sausage	Mycoprotein, soy protein	Mycoprotein, sunflower oil, ice, mixed spices, soy protein isolate, gluten, flour, salts	Physicochemical, microbial, nutritional, and mechanical assessments	Mixing	[66]
Meat analog	Soy protein, mycoprotein, oatmeal protein	Minced soy protein, mycoprotein, oatmeal protein, vegetable oil, tomato sauce	Cooking ability, food neophobia, sensory evaluation	Mixing	[67]
Nugget	Mycoprotein, texturized soy protein	Mycoprotein, onion, texturized soy protein, frying oil, salt, black pepper, wheat flour, batter ingredient (breading, sauce including pasteurized egg, salt, pepper)	Sensory evaluation, textural properties, color, proximate composition analyses	Mixing	[68]
Meat analog	Mycoprotein (*Neurospora intermedia*), soy protein isolate	Gluconolactone, mycoprotein (*Neurospora intermedia*), deionized water, soy protein isolate, soluble starch	Macroscopic, morphology, microstructural evaluation, FTIR, moisture, WHC, texture and mechanical properties	Mixing	[69]
Meat analog	Mycoprotein, potato protein	Mycoprotein, potato protein, distilled water, ferric pyrophosphate (FePP), sodium chloride, calcium chloride	Color, structure, cryo-SEM, energy dispersive spectroscopy, rheological evaluation, confocal laser scanning microscopy (CLSM), particle size measurement	Mixing	[70]
Meat analog	Soy protein, gluten, insect (black soldier fly larvae)	Water, sunflower oil, soy protein isolate, vital wheat gluten, insect (black soldier fly larvae)	Protein content, capillary rheometer, TPA	Mixing	[71]
Meat analog	Soy protein	Isolated soy protein, potato starch, calcium chloride, potassium chloride, xanthan gum, demineralized water, sodium alginate, carrageenan, glucomannan	Rheological analysis, 3D printing performance, CLSM, shrinkage, cooking loss, textural properties	3D printing	[72]
Meat analog	Soy protein isolate	Soy protein isolate, canola oil, ethyl cellulose, octenyl succinic anhydride starch, acetylated wheat starch, dodecenyl succinylated inulin, deionized water	Printing performance, microstructure, variable-pressure SEM, textural properties, protein solubility, thermal measurement, crystallography, oral tribology, temporal dominance of sensations	3D printing	[73]
Meat analog	Pea protein	Sodium alginate, sodium phosphate, distilled water, calcium chloride, pea protein isolate, transglutaminase (TGase)	Appearance, hardness, chewiness, springiness, adhesiveness, cohesiveness, cooking loss	3D printing	[74]
Meat analog	Soy protein isolate	Soy protein isolate, microcrystalline cellulose, citrate phosphate buffer, salt, beet juice extract, sunflower oil, water	Printing performance, morphology, dynamic sensory evaluation	3D printing	[75]
Meat analog	Pea protein	Pea protein isolate, starch, fat, soy lecithin, water	Rheological properties, forward extrusion testing, printing performance	3D printing	[76]
Meat analog	Mung bean protein	Mung bean protein isolate, beet red, distilled water, MC, xylose	Shear modulus, printing performance, visual appearance, instrumental color measurement, TPA, tensile test analysis, SEM, FTIR	3D printing	[77]
Meat analog	Mung bean protein	Mung bean protein isolate, MC, distilled water, TGase	Rheological properties, 3D printing performance, textural analysis, cooking conditions, cooking losses, shrinkage, microstructural analysis	3D printing	[78]
Meat analog	Texturized pea protein, single-cell protein	Texturized pea protein, single-cell protein, locust bean gum, sodium alginate, water	Rheological properties, microstructural properties, physical properties, maximum cutting force, maximum force required for 50% compression, anisotropy index, appearance	3D printing	[79]
Beef	Pea protein, soy protein, gluten, mushroom	Pea protein, soy protein, wheat protein, potato starch, maltodextrin, xanthan gum, salt, beetroot extract, coconut oil, water, mushroom (reishi, saffron milk cap, oyster)	Factors affecting 3D printing (nozzle height, printing speed, flow rate, height, width, length, number of layers), re-printability, rheological properties (strain sweep, thixotropy, frequency sweep), SEM, color characteristics, TPA, cooking loss, amino acid profile, sensory analysis	3D printing	[80]
Meat analog	Pea protein	Pea protein isolate, starch, fat, soy lecithin, water	Moisture content, protein content, TPA, microstructure, protein solubility	3D printing	[81]
Meat analog	Soy protein, chickpea protein, potato protein, canola protein	Soy protein isolate, chickpea protein isolate, potato protein isolate, canola protein isolate, gluten, distilled water, canola oil, MC, color, flavors, texturized vegetable protein	Amino acid composition, protein content, sodium dodecyl sulfate–polyacrylamide gel electrophoresis, surface hydrophobicity, particle charge detection, exposed and buried sulfhydryl groups, protein solubility, WAC, OAC, dynamic shear rheological properties, free water content, hardness, chewiness, gumminess	3D printing	[82]
Meat analog	Soy protein, wheat gluten, rice protein	Soy protein isolate, wheat gluten, rice protein, distilled water, canola oil	Rheological measurements, low-field nuclear magnetic resonance measurements, 3D-printing performance (height, surface, area), CLSM, TPA, SEM	3D printing	[83]
Meat analog	Mung bean protein isolate, wheat gluten	Mung bean protein isolate, wheat gluten, l-cysteine, distilled water	Printing characteristics, dimensional stability, fibrous structure formation, FTIR, WHC, SEM, texture profile analysis, electric tongue analysis, electric nose analysis	3D printing	[84]
Hamburger	Soy protein isolate, wheat gluten	Soy protein isolate, wheat gluten, insoluble dietary fiber (okara), beet red, deionized water	Printability assessment, WHC, texture property analysis, tensile test analysis, rheological properties, intermolecular forces determination, FTIR, SEM	3D printing	[85]
Meat analog	Soy protein, *Tenebrio molitor*	Freeze-dried *Tenebrio molitor*, soy protein isolate, distilled water	Proximate composition, crude protein content, fat content, moisture content, ash content, carbohydrate content, sodium dodecyl sulfate–polyacrylamide gel electrophoresis, attenuated total reflection–FTIR, rheological properties, 3D-printing performance, post-processing capacities, dimensional stability, TPA, field emission scanning electron microscopy	3D printing	[86]
Meat analog	Cricket (*Gryllus bimaculatus*), soy protein	Cricket fractions, soy protein isolate, distilled water	Moisture, ash, crude fat, crude protein content, sodium dodecyl sulfate-polyacrylamide gel electrophoresis, rheological analysis, three interval thixotropy test, post processing characteristics (dimensional stability, mechanical properties, field emission–scanning electron microscopy)	3D printing	[87]

**Table 4 foods-14-00498-t004:** Findings from 13 selected articles related to fish analogs.

Type of Analog	Protein Type	Composition	Tested Parameters	Technology	Reference
Fish ball	Soy protein isolate	Soy protein isolate, konjac glucomannan (KGM), deionized water, sea salt, sunflower oil, sucrose, sodium carbonate, ice	Moisture, protein, total fat, total ash, cooking yield, pH, TPA, rheological properties, SEM, FTIR	Mixing	[102]
Fish ball	Soy protein isolate	Soy protein isolate, KGM, sodium carbonate	Rheological properties, dynamic mechanical thermal analysis, determination of mechanical spectra, measurement for molecular interaction forces, powder X-ray diffraction, FTIR, CLSM	Mixing	[103]
Scallop	Pea protein isolate	Pea protein isolate (from yellow pea flour), TGase, citrus peel pectin	FTIR, TPA, WHC colorimetric analysis, cookability	Mixing	[104]
Sea foie gras analog	Duckweed RuBisCO protein	Duckweed RuBisCO protein, flaxseed oil, β-carotene, water	SEM, confocal, dynamic shear rheology, impact of heating conditions on textural properties, impact of droplet size on textural properties, appearance, colorimetry	Mixing	[105]
Fish ball	Soy protein isolate	KGM, soy protein isolate, deionized water, salt, sunflower oil, sucrose, dietary alkali, ice	Moisture content, fat content, temperature variation during frying, kinetic modelling of mass transfer and heat transfer, texture characterization during frying, color measurement, CLSM	Mixing	[106]
Scallop	Pea protein	Pea protein (from yellow pea flour) solution, citrus peel pectin, TGase (crosslinking enzyme), flaxseed oil emulsion, distilled water	In vitro digestion (INFOGEST), particle dimensions (laser diffraction) and surface charge (electrophoresis), CLSM, protein hydrolysis (pH-stat method)	Mixing	[107]
Fish cake	Brown rice protein isolate, pea protein isolate	Brown rice protein isolate, pea protein isolate, high acyl gellan gum, CG, MC, microbial TGase, canola oil, water, seasonings and flavors	Texture profile analysis, moisture content determination, expressible moisture and oil content, rheological properties, in vitro protein digestion, profile of released amino acids after in vitro protein digestion, confocal, sensory evaluation	Mixing	[108]
Fish ball	Macroalgae (nori, kombu)	Oyster mushroom, nori, kombu, corn flour, baking powder, salt, monosodium glutamate (water and all-purpose flour for the outside)	Ingredient optimization, antioxidant capacity, total flavonoid content, total phenolic content, chlorophyll, acid value, peroxide value, moisture content, total plate, yeast, and mold count	Mixing	[109]
Fish cake	Pea protein isolate, microalgae (*Nannochloropsis oceanica*)	Pea protein, *Nannochloropsis oceanica* (microalgae), microbial TGase, salt, monosodium glutamate, sugar, white pepper, sunflower oil, dextrose monohydrate, MC, gellan gum, cold water with ice	TPA, expressible moisture, oil content, rheological measurements, in vitro digestion, nuclear magnetic resonance spectroscopic analysis, spectral processing and analysis, microstructure test	Mixing	[110]
Fish analog	Soy protein isolate	Soy protein isolate, potato starch, MC, distilled water	Rheological properties, 3D-printing behavior, thermo-rheological behavior, extruded hardness, cutting strength, TPA	3D printing	[111]
Salmon fillet	Soy protein isolate	Isolated soy protein, annatto, saffron, white bean pulse flour, ultrapure water, soy oil	Phenolic compounds, voltammetric analysis, antioxidant capacity	3D printing	[112]
Fish analog	Soy protein isolate	Soy protein isolate, xanthan gum, rice starch, deionized water	Low-field nuclear magnetic resonance measurements, FTIR, SEM, TPA, printability	3D printing	[113]
Salmon fillet	Red lentil protein, yellow pea protein	Red lentil protein, deionized water, astaxanthin containing camelina oil, microbial TGase, pea protein extract	Rheological properties, TPA, measurement of extrudate thickness, CLSM	3D printing	[114]

## Data Availability

The raw data supporting the conclusions of this article will be made available by the authors on request.

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
