# Peer review of "Alternative Protein-Based Meat and Fish Analogs by Conventional and Novel Processing Technologies: A Systematic Review and Bibliometric Analysis"

_foods, 2025, doi:10.3390/foods14030498_

Round 1

Reviewer 1 Report

Comments and Suggestions for Authors

The systemic literature review on the meat and fish analog is well written. The language is easy to understand and clear. The hypothesis is well described.

I have the following suggestions to improve its readability and quality.  

·         Abstract: Well written

·         Keywords: may delete seafood analog and add extrusion

·         Introduction: L35: CO2 or GHG; please also quantify this, so to provide better insight

·         L52-53: please add a detail description on alternative protein with examples

·         L87- 88: need a suitable reference

·         L127: from ---to May 2024

·         L183: May 2024

·         Table 3 & 4: please may add more information/ data on the protein percentage/composition/processing parameters

·         Please add research gaps or your recommendations before the conclusion

Reviewer 2 Report

Comments and Suggestions for Authors

The manuscript deals with a current topic.
It is adequately designed. It handles cited references comprehensively. Sunflower seed cake is also used as protein for the production of meat analogs, but you only cite the protein of canola cake and canola. Please show quotes from articles that analyzed sunflower seed cake as well.
In lines 293 to 296, only the positive aspects of the technique of 3D printing of meat analogs are given. Please also mention the negative aspects, such as low scalability.
In presenting the production technology, please list both positive and negative aspects.
Serial numbers of the literature are duplicated.

Reviewer 3 Report

Comments and Suggestions for Authors

The manuscript ‘Alternative protein-based meat and fish analogs by conventional and novel processing technologies: A systematic review and bibliometric analysis’ by Gürbüz et al. has been reviewed.

The paper describes the research on developing fish and meat analogs from 2003 to 2023. Specifically, the authors focus on the proteins and the technology used.

The argument is of interest and I appreciated the paper.

Limits: The authors list a large number of studies and reviews, but only superficially discuss non-invasive approaches, in particular, few examples of marketed products are provided. To make the study more interesting to the readers authors should include a critical analysis of the approaches currently used, and a comparison among them, focusing more attention on potential advantages, disadvantages and costs.

The manuscript presents some imprecisions, and typos that should be revised before publication.

- The quality of the figures should be improved, as they stand are not readable (in particular Figures 1, 2, 11).
